# Genomic signatures of heterokaryosis in the oomycete pathogen *Bremia lactucae*

Kyle Fletcher[1], Juliana Gil[1,2], Lien D. Bertier[1], Aubrey Kenefick[1], Kelsey J. Wood[1,3], Lin Zhang[1], Sebastian Reyes-Chin-Wo[1,3,6], Keri Cavanaugh[1], Cayla Tsuchida[1,2,7], Joan Wong[1,4,8] & Richard Michelmore[1,5]

Lettuce downy mildew caused by *Bremia lactucae* is the most important disease of lettuce globally. This oomycete is highly variable and rapidly overcomes resistance genes and fungicides. The use of multiple read types results in a high-quality, near-chromosome-scale, consensus assembly. Flow cytometry plus resequencing of 30 field isolates, 37 sexual offspring, and 19 asexual derivatives from single multinucleate sporangia demonstrates a high incidence of heterokaryosis in *B. lactucae*. Heterokaryosis has phenotypic consequences on fitness that may include an increased sporulation rate and qualitative differences in virulence. Therefore, selection should be considered as acting on a population of nuclei within coenocytic mycelia. This provides evolutionary flexibility to the pathogen enabling rapid adaptation to different repertoires of host resistance genes and other challenges. The advantages of asexual persistence of heterokaryons may have been one of the drivers of selection that resulted in the loss of uninucleate zoospores in multiple downy mildews.

[1] Genome Center, University of California, Davis, CA 95616, USA. [2] Plant Pathology Graduate Group, University of California, Davis, CA 95616, USA. [3] Integrated Genetics and Genomics Graduate Group, University of California, Davis, CA 95616, USA. [4] Plant Biology Graduate Group, University of California, Davis, CA 95616, USA. [5] Departments of Plant Sciences, Molecular and Cellular Biology, Medical Microbiology and Immunology, University of California, Davis, CA 95616, USA. [6] Present address: Bayer Crop Science, 37437 CA-16, Woodland, CA 95695, USA. [7] Present address: Arcadia Biosciences, Davis, CA 95616, USA. [8] Present address: Pacific Biosciences of California, Inc., Menlo Park, CA 94025, USA. Correspondence and requests for materials should be addressed to R.M. (email: rwmichelmore@ucdavis.edu)

O omycetes are genetically and biochemically distinct from fungi[1] but have similar infection strategies and architectures. Oomycetes include successful and diverse plant and animal pathogens with global economic impacts[2,3]. These include the downy mildews caused by biotrophic members of the Peronosporaceae that are challenging to study due to their obligate reliance on their host. These pathogens are highly variable; plant resistance genes and fungicide treatments are often rapidly overcome[4–7]. Various mechanisms have been proposed for rapid generation of genetic diversity including hyper-mutability of genomic regions that encode effectors[8], changes in ploidy[9,10], and parasexuality[11].

Heterokaryosis, the state of having multiple genetically distinct nuclei in a single cell, is an important life history trait in some true fungi[12]. While transient heterokaryosis has been suggested and detected in oomycetes[5,10,13–15], the impacts of heterokaryosis remain poorly understood and rarely considered. The life cycles of many oomycetes are not conducive to the propagation of stable heterokaryons because most produce multiple flagellated, mononucleic, motile spores from sporangia[15,16]; heterokaryons are consequently broken every asexual generation[15]. However, some downy mildew species including *Bremia lactucae* do not produce zoospores and germinate directly from multinucleate sporangia[17,18], transmitting multiple, possibly genetically distinct nuclei in each asexual generation.

*Bremia lactucae* is an obligate biotroph that causes lettuce downy mildew, the most important disease of lettuce worldwide. Numerous races and population shifts have been documented in Europe, Australia, Brazil, and California[19–21]. Resistance genes are rarely durable in the field and curative fungicides have become ineffective[4–7]. Several mechanisms for variation have been documented. *B. lactucae* is predominantly heterothallic with two mating types and sexual reproduction can generate new virulence phenotypes[22]. Asexual variation also occurs but is less well understood. Somatic fusion resulting in either polyploids or heterokaryons has been observed[5,14], but it remained unclear whether heterokaryosis or polyploidy are significant sources of stable phenotypic variation of *B. lactucae*. Previously, sexual progeny of *B. lactucae* have been generated to build genetic maps[23,24], study the genetics of (a)virulence and metalaxyl insensitivity[4,6,25,] and infer the presence of accessory chromosomes[26]. Only limited genomic studies had been conducted due to the difficulties of studying this biotrophic species[27].

This study presents a near-chromosome-scale genome assembly of *B. lactucae* using multiple sequencing technologies and assembly approaches. This resource, combined with genome size estimates generated by flow cytometry, demonstrates the prevalence of heterokaryosis in multiple *B. lactucae* isolates and the absence of polyploidy. Heterokaryons are shown to be somatically stable and fitter on non-selective hosts compared with homokaryotic derivatives. Homokaryotic components differ in (a)virulence phenotypes and confer viability on selective hosts. Selection should be considered as acting on a population of nuclei within a coenocytic mycelium to maximize somatic hybrid vigor.

## Results

**Genome assembly**. *Bremia lactucae* isolate SF5 was initially assembled into 885 scaffolds over 1 Kb with a contig $N_{50}$ of 30.6 Kb and a scaffold $N_{50}$ of 283.7 Kb. The haploid genome size of this isolate and 38 others were estimated to be ~152 Mb (+/−3 Mb) by flow cytometry (Fig. 1a and Supplementary Table 1). This 115 Mb assembly contained 91 Mb of sequence plus 24 Mb of gaps. Subsequently, 87.9 Mb (96.5%) of the assembled sequence was placed into 22 scaffolds over 1 Mb using Hi-C; these totaled 112 Mb including gaps. The resultant

assembly was highly collinear and comparable with the highly contiguous v3.0 assembly of *Phytophthora sojae*[28], which cross-validates the high quality of both assemblies (Fig. 1b). The heterozygosity of isolate SF5 was 1.17% and ranged from 0.77% to 1.29% for other isolates. These levels ranked high compared with other oomycetes, the majority of which had less than 1% heterozygosity (Fig. 1c). This level of heterozygosity resulted in some alleles not being collapsed into a consensus sequence, necessitating multiple rounds of condensation to achieve a close to haploid consensus assembly.

The discrepancy between the final assembly size (91 Mb without gaps) and genome size measured by flow cytometry (152 Mb) is due to collapsed repeats in the assembly. Single-copy k-mers were present in the predicted proportions in the assembly; most of the homozygous k-mers and approximately half of the heterozygous k-mers were distributed across the two peaks as expected (Fig. 1d). BUSCO[29] analysis with the protist database (v9) also revealed 98.3% completeness similar to other well-assembled oomycetes (Supplementary Table 2). Therefore, the assembly contains most of the single-copy portion of the genome. Repeat annotation followed by masking determined that 63 Mb of the 91 Mb assembled sequence was repetitive (Table 1). The majority of the annotated repeats were recently diverged long terminal repeat retrotransposons (LTR-RTs). Annotation identified 6.3 Mb as *Copia* (RLC) and 53.3 Mb as *Gypsy* (RLG) elements (Table 1). The average coverage of sequences annotated as repeats in the assembly was 2.1-fold higher than that of the annotated genes. Therefore, the 63 Mb repeat portion of the assembly is present at least twice in the haploid genome accounting for the 61 Mb difference between the assembly and the genome size determined by flow cytometry.

Divergence of long terminal repeat (LTR) pairs showed that the majority of these repeat elements were recently expanded (Supplementary Fig. 1), when compared with previously published downy mildews (Fig. 2b). *B. lactucae* had a high density of recently diverged LTRs similar to *Phytophthora* spp. (Fig. 2a). This suggests that *B. lactucae* and *Phytophthora* species surveyed have undergone recent expansions of *Copia* and *Gypsy* elements. The larger genome assemblies of both *S. graminicola* isolates contained the largest number of annotated LTR-RTs of any downy mildew surveyed (Fig. 2b), although LTR pairs were more diverged than in *B. lactucae* (Fig. 2a). Significantly, when LTR-RTs from each species were used to mask the assemblies, *B. lactucae* contained the highest proportion of LTR-RTs with 74.2% of its contig sequence masked (Fig. 2c). In most of the other species studied, 26 to 46% of contig sequences were masked, except for the larger assemblies of *S. graminicola* (two isolates) and *P. infestans* that had 72.1%, 70.7%, and 62.6% masked, respectively (Fig. 2 c). The high frequency of low-divergence repeat sequences in *B. lactucae* combined with high heterozygosity (Fig. 1c) may have confounded assembly algorithms and slowed the generation of an accurate assembly, as well as prevented the construction of whole chromosomal molecules. Interestingly, LTR divergence of *Plasmopara viticola*, which is a closer relative to *B. lactucae* than any *Phytophthora* spp. (Fig. 3; see below), does not have the same recent expansion of LTRs; this implies that this expansion of LTR-RTs was not ancestral to these species.

**Phylogenomics**. Phylogenetic analysis of 18 proteins identified with BUSCO[29] that were ubiquitously single-copy in the species surveyed supported polyphyly of downy mildew species within the nine *Phytophthora* clades analyzed (Fig. 3). *B. lactucae* clustered with the two *Plasmopara* spp. This clade was closely associated with *Phytophthora* clade 1, which includes *P. infestans* and *P. cactorum*. *B. lactucae* did not cluster with *Peronospora effusa*,

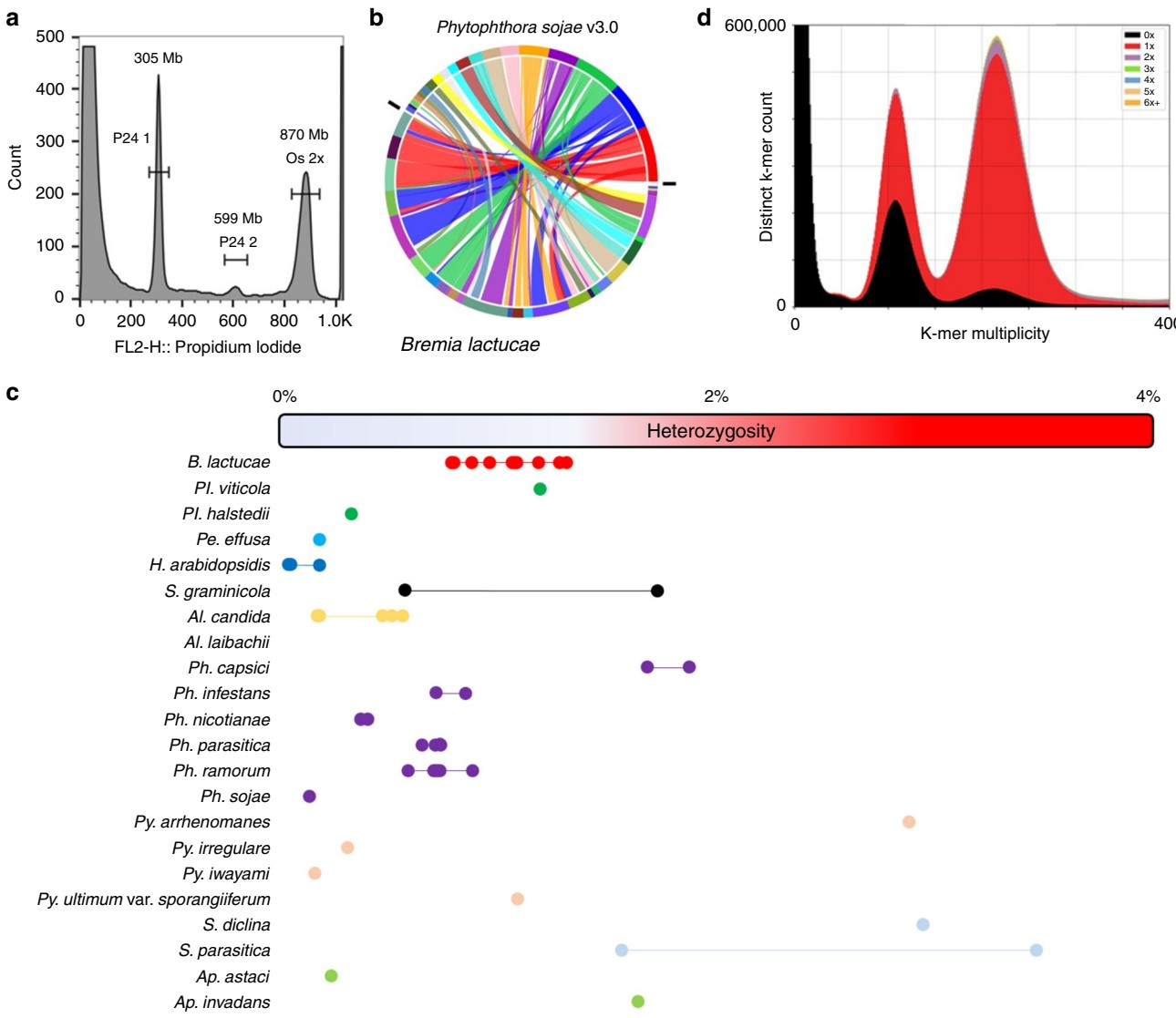

**Fig. 1** Genome and assembly features of *B. lactucae*. **a** Estimation of genome size of heterokaryotic isolate C82P24 by flow cytometry. The nuclei of *B. lactucae* have two peaks calibrated relative to the reference nuclei of *Oryza sativa* (Os, 2 C = 867 Mb). Nuclei of isolate C82P24 (P24) were estimated to be 305 Mb (P24 1, 2 C) and 599 Mb (P24 2, 4 C). Another 38 isolates all have similar sizes (Supplementary Table 1). **b** Extensive collinearity between *B. lactucae* and *P. sojae* displayed as a SyMap plot. **c** Comparison of heterozygosity in 54 isolates of 22 oomycete species (Supplementary Table 9). Dots, representative of a single isolate, are joined by bars to aid interpretations of each species. **d** High quality of *B. lactucae* assembly demonstrated by inclusion of k-mers from paired-end reads in the assembly. Colors indicate presence of k-mers in the assembly, relative to reads. Black: the distribution of k-mers present in the read set but absent in the assembly. Red: K-mers present in the read set and once in the assembly. Purple: K-mers present in the read set and twice in the assembly. The first peak depicts heterozygous k-mers and the second peak depicts homozygous k-mers. A high-quality consensus assembly will contain half the k-mers in the first peak, the other half of which should be black due to heterozygosity, and all the k-mers in the second peak should be present only once, which therefore should be red. Very few duplicated k-mers were detected in the SF5 assembly. K-mers derived from repeat sequences have higher multiplicity and are not plotted. Source data for panel c is provided in the Source Data file

### Table 1 Repeat statistics of the *B. lactucae* assembly

|  | Number of elements | Total length (bp) | Percentage of contig sequence |
|---|---|---|---|
| Long terminal repeat elements | 63,720 | 61,227,642 | 67.3% |
| *Copia* | 5659 | 6,314,733 | 6.9% |
| *Gypsy* | 57,655 | 53,338,585 | 58.6% |
| Short interspersed nuclear element | 35 | 15,270 | 0.02% |
| Long interspersed nuclear repeat | 182 | 471,735 | 0.52% |
| DNA elements | 337 | 471,455 | 0.52% |
| Unclassified | 685 | 1,050,820 | 1.15% |

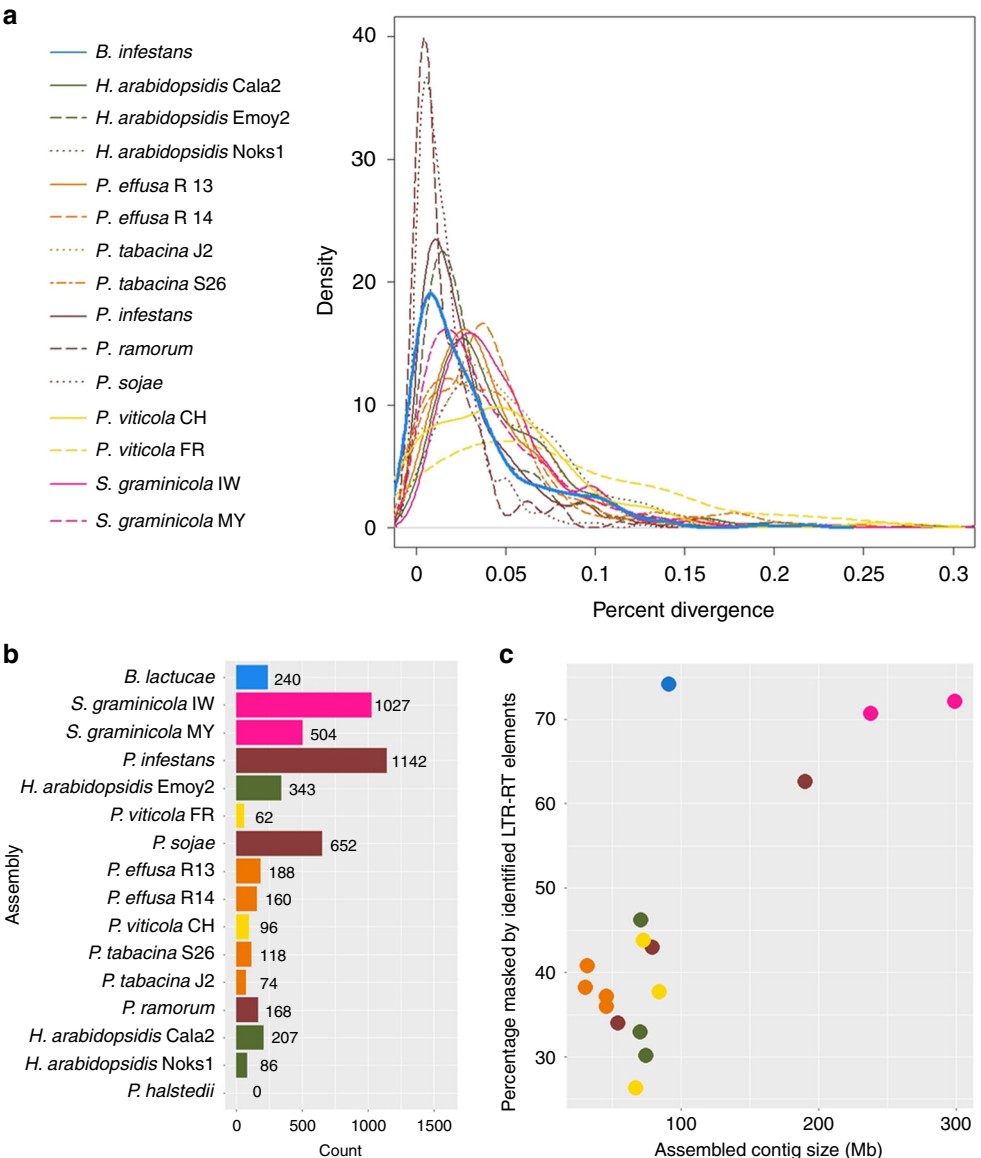

**Fig. 2** Comparative long terminal repeat retrotransposon (LTR-RT) analysis. **a** Comparison of percent divergence of LTR elements in 15 oomycete assemblies. Distribution of percent divergence of LTR elements is shown for 12 downy mildew (*B. lactucae*, *H. arabidopsidis*, *P. effusa*, *P. tabacina*, *P. viticola*, and *S. graminicola*) and three *Phytophthora* (*P. infestans*, *P. ramorum*, and *P. sojae*) assemblies. Statistics of these assemblies are included in Supplementary Table 2. LTR elements of *B. lactucae* are less diverged than elements in other downy mildew assemblies. **b** Counts of unique LTR-RTs harvested and annotated from each assembly surveyed. Larger assemblies (panel **c**) are observed as having higher counts of LTR-RTs. Bars are ordered by the percent of the assembly masked displayed in panel **c**. Only partial elements could be found for *P. halstedii*. **c** Scatterplot demonstrating the percentage of the assembly sequence that is masked by annotated LTR-RTs and partial elements. Colors are retained from panel **b**. The percentage of the assembly masked increases with assembly size. *B. lactucae* is an outlier as it has a medium assembly size, but the highest masked percentage. Source data for all panels are provided in the Source Data file

*Pseudoperonospora cubensis*, *Sclerospora graminicola*, and *Hyaloperonospora arabidopsidis*. This second downy mildew clade clustered closer to *P. agathidicida* in *Phytophthora* clade 5. This is consistent with the biotrophic downy mildews having evolved at least twice from hemi-biotrophic *Phytophthora*-like ancestors. These results are similar to previous, less extensive studies[17,30]. Additional genome sequencing of both biotrophic and hemi-biotrophic Peronosporales species will enable further clarification with regards to the origin(s) of the downy mildews.

**Gene annotation**. Ab initio annotation identified 9781 protein-encoding genes. More than half (5009) lacked introns, 1949 had one intron, 1063 had two introns, and 1760 had three or more introns. A maximum of 20 introns was observed in two genes. The average gene length was 1679 bp and ranged from 180 bp to 20.7 Kb. The 142 largest genes were conserved with other oomycete gene models. In total, orthologs were identified for 84% of the predicted models. The mean exon length was 664 bp, ranging from 4 bp to 18.3 Kb, while the mean intron length was 104 bp, ranging from 10 bp to 9.3 Kb. Strand specific transcriptional support was detected for 74% of the genes. The total gene space was 16.5 Mb, of which 15.2 Mb was exonic and 1.5 Mb was intronic. This is similar to other obligate biotrophic oomycetes, where the gene space ranges from 13.5 to 25.8 Mb. The hemi-biotrophic *P. sojae* has the largest reported gene space within the Peronosporaceae at 37.7 Mb (Supplementary Table 3).

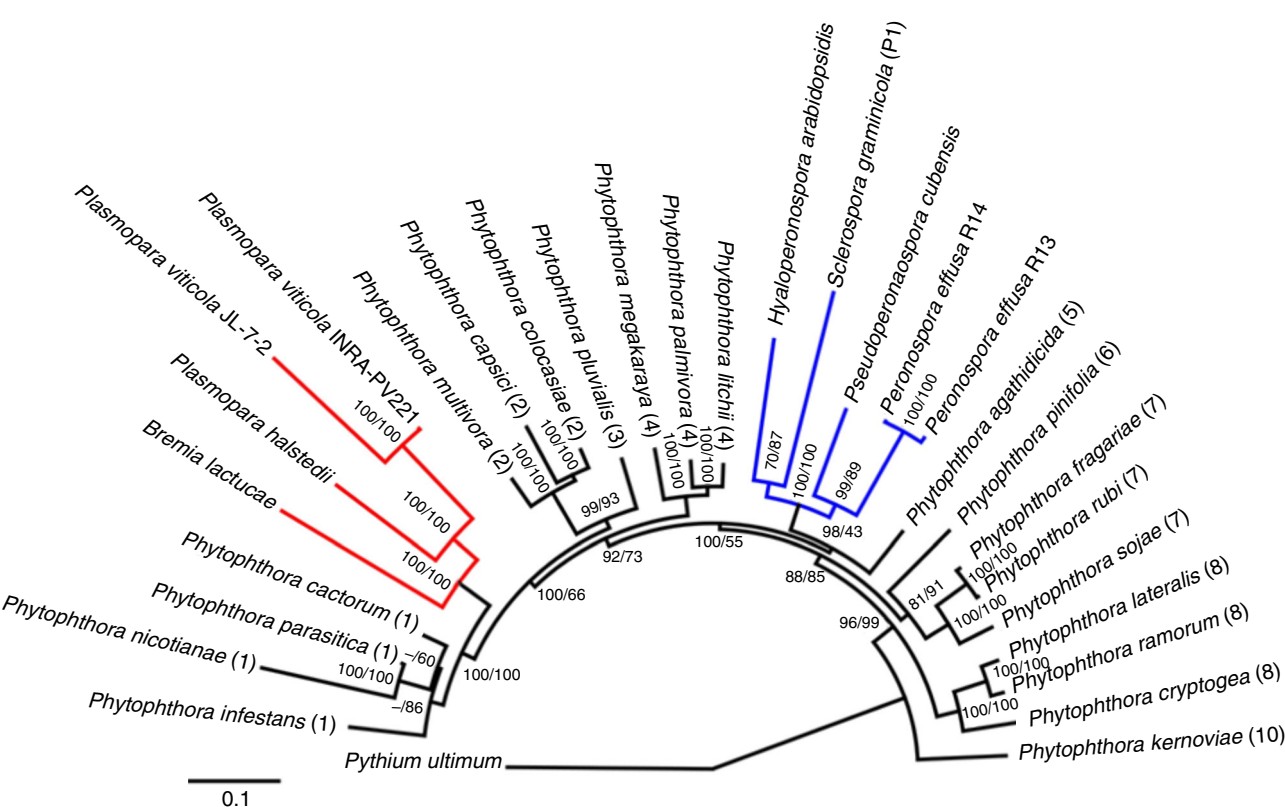

**Fig. 3** Polyphyly of downy mildews and paraphyly of *Phytophthora* spp. Phylogenetic maximum likelihood tree based on the analysis of 18 BUSCO protein sequences across 29 Peronosporaceae species rooted with *Pythium ultimum* as the outgroup. The 20 *Phytophthora* species were selected to represent assemblies from the nine published *Phytophthora* clades indicated by the number in brackets. No assembly from *Phytophthora* clade 9 was available. Downy mildew clades 1 and 2 are shown in red and blue, respectively. Support for nodes is shown as percent bootstrap values from 1000 iterations of the nucleotide/protein alignments. The output nucleotide and protein trees only disagreed in the order of *Phytophthora* clade 1 species and are indicated by -- nucleotide support. Scale is the mean number of amino acid substitutions per site. Nucleotide and protein decatenated alignments and output trees are provided as Source Data

Motif searches revealed the repertoire of candidate effector proteins that could be important in pathogenesis. Among a total of 159 candidate secreted RxLR effectors, 66 had a canonical RxLR motif and 93 had degenerate [GHQ]xLR or RxL[GKQ] motifs[17,27,31]; 63 candidates also encoded a [DE][DE][KR] motif[32] and/or a WY domain[33,34] (Table 2). Expression inferred by presence in the transcriptome assembly was detected for 109 of these candidate RxLR effectors, 35 of which also had an EER motif or WY domain (Table 2). In addition to the 161 RxLR-EER candidates, there were 24 secreted proteins and 23 proteins lacking a secretion signal predicted with one or more WY domains but no detectable RxLR motif. Of these, 19 of the 24 WY proteins lacking an identifiable RxLR motif with a signal peptide and 13 of the 23 without a signal peptide were detected in the transcriptome (Table 2). Interestingly, an EER or EER-like motif was detected in the first 100 residues from 29 of the 45 WY proteins that lacked an RxLR motif, 20 of which were predicted to be secreted. This is consistent with not all effectors requiring an RxLR motif for translocation in to the host cell, similar to previously reported effectors in animal pathogenic oomycetes[35]. Two putative secreted Crinklers (CRNs)[36,37] were annotated, one of which also contained an RxLQ and DDR motif. An additional 74 CRNs lacking a secretion signal were identified, although only six of these were present in the transcriptome assembly (Table 2). Four of these six had the canonical LFLAK motif and the other two had a LYLA motif[36,37]. Together, these candidate effectors comprise 1.9% of all genes annotated in *B. lactucae*. Orthologs of all proteins that have previously been described as inducing a host

**Table 2 Counts of annotated effectors in the *B. lactucae* genome and transcriptome assemblies**

| | Genome | Transcriptome[a] |
|---|---|---|
| RxLR | 35 | 27 |
| [GHQ]xLR | 31 | 20 |
| RxL[GKQ] | 30 | 27 |
| RxLR-EER | 22 | 13 |
| [GHQ]xLR-EER | 18 | 8 |
| RxL[GKQ]-EER | 11 | 4 |
| RxLR-WY | 3 | 1 |
| RxL[GKQ]-WY | 2 | 2 |
| RxLR-EER-WY | 6 | 6 |
| [GHQ]xLR-EER-WY | 1 | 1 |
| SP-WY | 24 | 19 |
| WY | 23 | 13 |
| SP-CRN | 2 | 0 |
| CRN | 74 | 6 |
| Total proteins with RxLR | 66 | 47 |
| Total proteins with degenerate RxLR | 93 | 62 |
| Total proteins with WY domain | 59 | 42 |
| Total Crinklers | 76 | 6 |

[a]Presence in transcriptome inferred by tBLASTn

response[38–40] were detected in the draft assembly (Supplementary Table 4). An additional 173 proteins (1.8% of all annotated genes) had domains ascribed to putative pathogenic functions in studies of other species (Supplementary Table 5). This is lower than the

proportion reported for *Phytophthora* spp. (2.6 to 3.6%) and consistent with observations for other downy mildews where 1.3 to 1.7% of total annotated proteins had putative pathogenicity domains[17].

The majority of genes encoding flagella-associated proteins and calcium-associated domains were missing from the *B. lactucae* assembly. *B. lactucae* has lost 55 of 78 orthogroups that contain flagellar proteins (Supplementary Fig. 2). One hundred and twelve proteins from *P. infestans* were present in these orthogroups; 78 of these proteins were absent in *B. lactucae*. This is similar to assemblies of other non-flagellate downy mildews that had 34 to 48 proteins in these orthogroups (Supplementary Fig. 2). This is consistent with the loss of zoospore production by *B. lactucae*. There was also a significant loss of calcium-associated domains, which is also observed in the assemblies of other non-flagellate downy mildews[17]. *B. lactucae* had no proteins present in 125 of the 177 calcium-associated orthogroups similar to other non-flagellates, which ranged from 118 to 125. These orthogroups contained 53 proteins from *B. lactucae* compared with 193 proteins in *P. infestans*. Other non-flagellate species had 52 to 59 proteins assigned to these orthogroups (Supplementary Fig. 2). The parallel loss of zoospore production and proteins with calcium-associated domains in both clades of downy mildews (Fig. 3) is consistent with the involvement of these proteins in zoosporogenesis[41]. Genes encoding carbohydrate binding, transporter, and pathogenicity associated domains were also under-represented in *B. lactucae*, as previously reported for other downy mildews in both clades[17]. This provided further evidence for the convergent loss of genes encoding these domains during adaptation to biotrophy.

The majority of annotated genes had levels of coverage close to the average sequencing depth (Supplementary Fig. 3), indicating that most genes were each assembled into a single consensus sequence. A minority of genes had a normalized read depth equal to half the sequencing coverage, consistent with divergent haplotypes that had assembled as independent sequences. The BUSCO[29] genes had the same distribution (Supplementary Fig. 3). However, genes encoding candidate effectors had variable coverage (Supplementary Fig. 3); this could have been due to a disproportionate number of effector haplotypes being assembled independently and/or a high rate of divergence between haplotypes resulting in poor mapping rates.

**Genomic signatures of heterokaryosis.** Distinct alternative allele frequency profiles were detected in multiple isolates of *B. lactucae* (Fig. 4 and Supplementary Fig. 4). Such analysis had previously been used to support polyploidy in *P. infestans*[9,42]. The profiles of thirteen isolates, including the reference isolate SF5, were clearly unimodal, seven isolates were trimodal, and nine isolates were bimodal (Supplementary Fig. 4). Two other isolates had profiles that were not clearly bimodal or trimodal (Supplementary Fig. 4). The symmetrical unimodal distribution of SF5 was consistent with a diploid genome; the other distributions were not. However, the genome size for all isolates as measured by flow cytometry varied by less than 3%. In the case of polyploidy, the genome size of triploids and tetraploids would be 150% and 200% that of the diploid, respectively; therefore, there was no evidence for polyploidy in *B. lactucae* (Fig. 1a and Supplementary Table 1).

Further evidence against polyploidy was provided by analysis of sexual progeny from the segregating $F_1$ population of SF5 (unimodal; Fig. 4a) × California isolate C82P24 (trimodal; Fig. 4b)[14,23]. Four progeny isolates sequenced to over 50× all had unimodal allele frequency plots (Supplementary Fig. 5). Flow cytometry of 16 progeny isolates estimated the genome sizes to be the same as all other isolates (Fig. 1a and Supplementary Table 1).

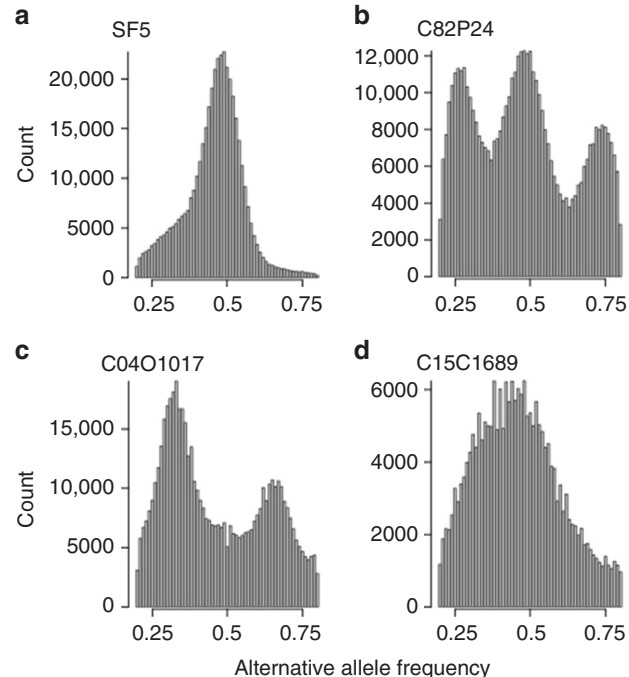

**Fig. 4** Heterokaryosis in *B. lactucae*. Example alternative allele frequency plots of SNPs detected in four field isolates of *B. lactucae*. **a** A unimodal distribution with a 1:1 ratio of reads supporting alternative and reference alleles seen in the homokaryotic SF5 isolate. **b** A trimodal distribution with peaks at 1:1, 1:3, and 3:1 ratios of reads supporting alternate alleles in the heterokaryotic C82P24 isolate, consistent with two nuclei being present in approximately equal proportions. **c** A bimodal distribution with two peaks at 1:2 and 2:1 ratios of reads supporting alternative alleles observed in the heterokaryotic isolate C04O1017, consistent with three nuclei being present in equal proportions. **d** The complex distribution observed in isolate C15C1689, consistent with an uneven mixture of multiple nuclei in a heterokaryotic isolate. Allele distributions of 31 isolates are shown in Supplementary Fig. 4. Sequencing statistics for these isolates are provided in Supplementary Table 8

The outcross origin of these progeny was confirmed by the presence of unique combinations of SNPs inherited from each parent. Therefore, these progeny isolates could not have arisen by apomixis or selfing and all sexual progeny from this unimodal × trimodal cross were diploid. The origins of the gametes in this cross were determined for 38 progeny isolates that had been sequenced to sufficient depth. Pairwise SNP-based kinship coefficients revealed two distinct half-sib families of 29 and 9 individuals (Fig. 5). Therefore, three rather than two nuclei contributed gametes in this cross. The trimodal alternative allele frequency plot and flow cytometry of C82P24 are consistent with this isolate being heterokaryotic with two diploid nuclei.

To confirm that C82P24 was heterokaryotic rather than a mixture of two isolates, 20 asexual derivatives were generated from single sporangia. Kinship of these 20 isolates was as high between one another as with the original isolate, indicating they were identical. Furthermore, all asexual derivatives of C82P24 displayed similar relatedness to all sexual progeny (Fig. 5). Sequencing of 11 asexual derivatives to >50x coverage demonstrated that they retained the trimodal profile, indicating that two distinct nuclei were present in each derivative (Supplementary Fig. 6) with a diploid size of 303 +/− 3 Mb as measured by flow cytometry (Supplementary Table 6). Therefore, C82P24 was heterokaryotic rather than a mixture of isolates.

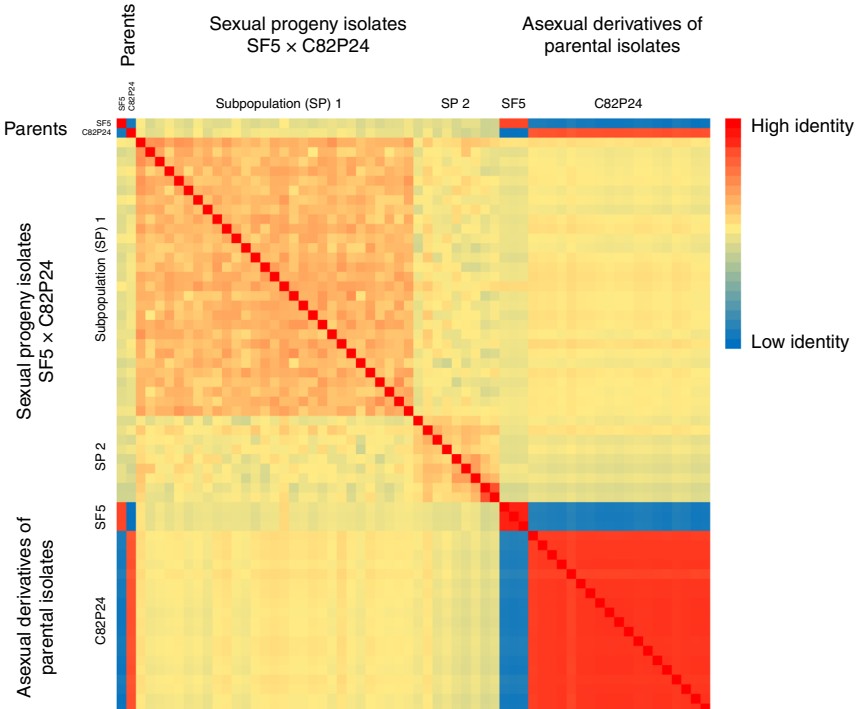

**Fig. 5** Sexual progeny from a cross between homokaryotic and heterokaryotic isolates form two half-sib groups. Kinship analysis based on SNPs segregating in sexual progeny generated by crossing SF5 (homokaryotic) with C82P24 (heterokaryotic). The first cluster delineates the majority of the offspring as one group of siblings derived from the same two parental nuclei (Subpopulation 1, SP 1). The second cluster delineates the remaining offspring as a second group of siblings derived from a different nucleus in C82P24 (Subpopulation 2, SP2). Relatedness of these two groups is consistent with having one parental nucleus in common derived from SF5. Relatedness of single-spore asexual derivatives of both isolates is also shown. Single-spore derivatives of C82P24 had a high relatedness to all other C82P24 derivatives and the original isolate. These derivatives and C82P24 were equidistant to all offspring, indicating the heterokaryotic C82P24 isolate had not been separated into homokaryotic components by generating single-spore derivatives. Source data is provided in the Source Data file

To demonstrate heterokaryosis in another isolate, 10 asexual derivatives were also generated from isolate C98O622b that also displayed a trimodal alternative allele frequency (Supplementary Fig. 4). In this case, kinship analysis revealed three distinct groups of derivatives (Fig. 6a). Derivatives A to F had a high kinship and an identical virulence phenotype to C98O622b (Fig. 6a, b). Derivatives G to I, formed a second group and derivative J a third; these four derivatives had lower kinship to C98O622b than derivatives A to F. The lowest kinship was between derivatives G to I and J (Fig. 6a). Virulence phenotypes varied between but not within groups (Fig. 6b); C98O622b and derivatives A to F were virulent on both *Dm4* and *Dm15*; derivatives G to I were avirulent on *Dm4* and virulent on *Dm15*, while derivative J was conversely virulent on *Dm4* and avirulent on *Dm15* (Fig. 6b). The single-spore derivatives of C98O622b were sequenced to >50× coverage to determine their nuclear composition. Derivatives A to F were trimodal (Fig. 6c, Supplementary Fig. 7); all other derivatives were unimodal, which is consistent with the separation of the heterokaryon into its diploid components (Fig. 6c). This conclusion was supported by combining read sets in silico. Combining reads of derivatives G-I did not increase their relatedness to C98O622b, while combining reads of derivatives G, H, or I with those of J resulted in a high kinship to C98O622b (Fig. 6a) and trimodal profile similar to C98O622b (Supplementary Fig. 8). Therefore, C98O622b was also heterokaryotic; however, unlike C82P24, C98O622b was unstable and could be separated into constituent homokaryotic derivatives by sub-culturing from single sporangia.

The trimodal distributions of the derivatives A to F were not identical and could clearly be split into two configurations,

suggesting different nuclear compositions of these derivatives. Derivatives B and F were similar to C98O622b, displaying peaks at approximately 0.25, 0.5, and 0.75 (Fig. 6c). The other four heterokaryotic derivatives A, C, D, and E had peaks at approximately 0.33, 0.5, and 0.67 (Fig. 6c). The nuclear composition of these heterokaryotic derivatives was investigated by subsampling SNPs identified as unique to each homokaryotic derivative (G to J). This revealed that in the trimodal distribution of derivatives B and F (peaks at 0.25, 0.5, 0.75) SNPs unique to either constituent nucleus were in peaks at 0.25 and 0.75 (Fig. 6d, Supplementary Fig. 9), consistent with a balanced 1:1 ratio of constituent nuclei (1:3 read ratio of SNPs). For derivatives A, C, D, and E (peaks at 0.33, 0.5, 0.67), SNPs unique to constituent nuclei resembling derivatives G to I were consistently in peaks at approximately 0.17 and 0.83, while SNPs identified as unique to derivative J were consistently in peaks at 0.33 and 0.67 (Fig. 6d, Supplementary Fig. 9), consistent with a 2:1 unbalanced nuclear ratio in favor of nuclei similar to derivative J. This was further supported by combining reads in silico. Combining reads from derivatives G, H, or I with J in equal proportions resulted in trimodal plots similar to those of derivatives B and F (peaks at 0.25, 0.5, and 0.75; Supplementary Fig. 8). Combining reads from derivatives G, H, or I with J in a ratio of 1:2 resulted in frequency profiles like those of derivatives A, C, D, and E (peaks at 0.33, 0.5, and 0.67; Supplementary Fig. 8). This supports an unequal nuclear composition in four of the asexual derivatives of C98O622b.

**Heterokaryosis impacts the rate of sporulation.** To investigate the potential benefits of heterokaryosis, the fitness of asexual

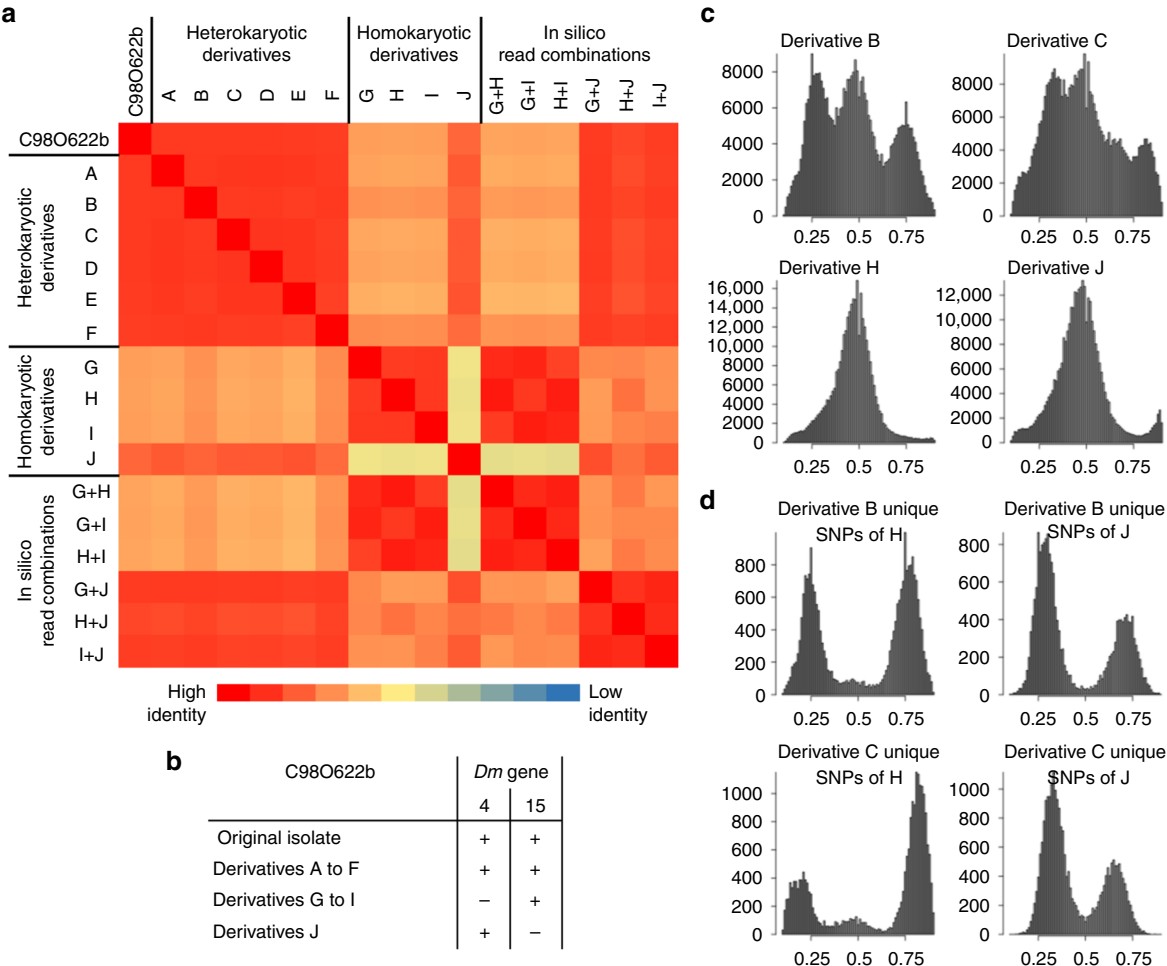

**Fig. 6** Genomic and phenotypic instability of the heterokaryotic isolate C98O622b. **a** Relatedness analysis of ten asexual single-spore derivatives of C98O622b placed them into three genomic groups. One group of derivatives, A to F, were heterokaryotic and highly similar to C98O622b. The other two groups, derivatives G to I and derivative J, were each homokaryotic, less similar to C98O622b than the heterokaryotic group was, and even less similar to each other. Combining reads in silico of isolates G to I did not change their relatedness to other isolates; combining reads of any of G to I with J scored similarly high in relatedness to C98O622b as derivatives A to F. **b** Phenotypic differences between heterokaryotic and homokaryotic derivatives of C98O622b compared with the original isolate. Derivatives A to F were virulent on both *Dm4* and *Dm15*; however, derivatives G to I were avirulent on *Dm4* and virulent on *Dm15*, while derivative J showed the reverse virulence phenotype. **c** Alternative allele frequency plots of four C98O622b derivatives showing that derivatives A to F are heterokaryotic and G to J are homokaryotic. Alternative allele frequency plots of the derivatives nine derivatives are shown in Supplementary Fig. 7. Derivative G is not presented as inadequate coverage was obtained (Supplementary Table 8). **d** Alternative allele frequency plots of heterokaryotic derivatives based only on SNPs unique to each homokaryotic derivative. In a balanced heterokaryon such as derivative B, SNPs unique to each homokaryon are observed at frequencies of 0.25 and 0.75, consistent with the presence of each nucleus in a 1:1 ratio. In an unbalanced heterokaryon, such as derivative C, SNPs unique to homokaryotic derivatives G, H, and I are present at frequencies of approximately 0.17 and 0.83, while SNPs unique to derivative J are present at frequencies of 0.33 and 0.66; this is consistent with twice as many nuclei of J as those of G, H, and I. Similar distributions are observed for derivatives A, D, and E, indicating that they are unbalanced heterokaryons (Supplementary Figs. 8, 9). Source data for panel a is provided in the Source Data file

derivatives of C98O622b was assessed on a universally susceptible cultivar and two differential host lines. Derivatives A and B were selected to represent unbalanced and balanced heterokaryons, respectively, while derivatives I and J represented the two homokaryons. When grown in four replicates on the universally susceptible lettuce cultivar (cv.) Green Towers, the heterokaryotic derivatives grew faster than either homokaryotic derivative (Fig. 7a). The balanced heterokaryotic derivative B was significantly fitter than the homokaryotic derivative I. There was no significant difference within heterokaryotic derivatives or within homokaryotic derivatives when grown on cv. Green Towers. This is consistent with heterokaryotic isolates being fitter when unchallenged by host resistance genes. However, when a product

of either nucleus of the heterokaryon was detected by a resistance gene (i.e., *Dm4* in R4T57D or *Dm15* in NumDM15) that differentiates the homokaryotic derivatives (Fig. 6b), the heterokaryotic derivatives were less vigorous than the virulent homokaryotic derivative (Fig. 7b). This suggested that it may be possible to break a heterokaryon by repeated subculture on a selective cultivar, as reported previously[5]. When the heterokaryotic derivatives were inoculated onto an $F_1$ hybrid of the selective lines expressing both *Dm4* and *Dm15*, neither the heterokaryotic nor homokaryotic derivatives were able to grow. Therefore, combining multiple resistance genes against the entire *B. lactucae* population into a single cultivar remains a potentially effective strategy to provide more durable resistance to the pathogen.

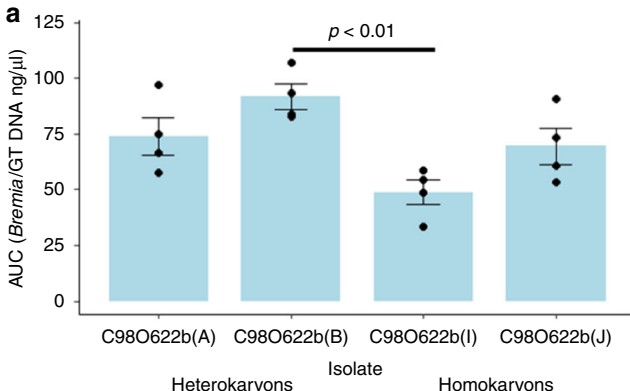

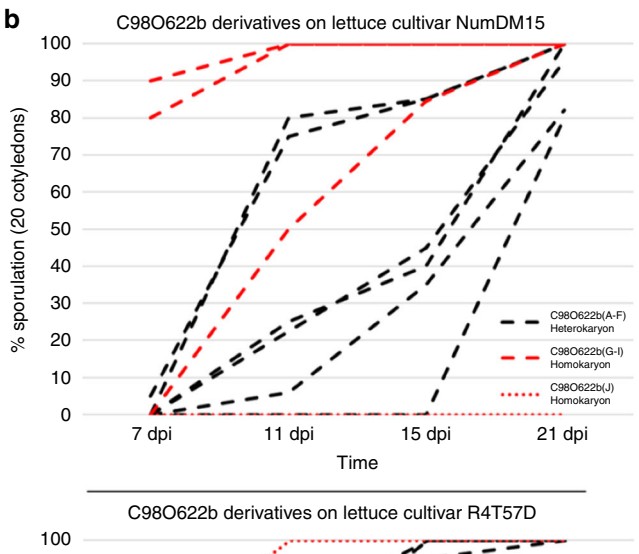

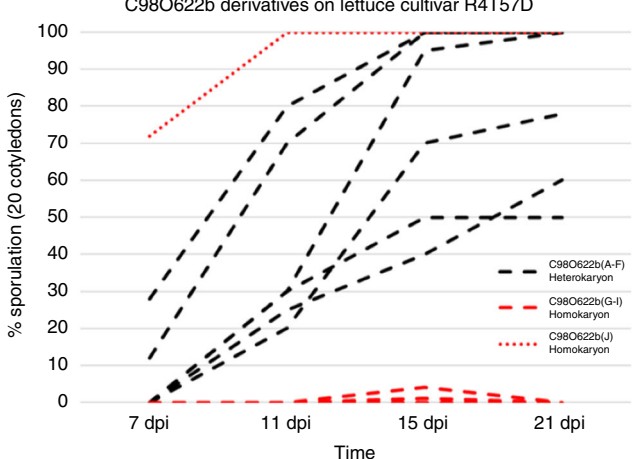

**Fig. 7** Differences in fitness between heterokaryotic and homokaryotic derivatives of C98O622b. **a** Growth of four single-spore derivatives on the universally susceptible lettuce cv. Green Towers (GT). Heterokaryons exhibit higher growth mass per lettuce seedling and DNA quantity collected per ml of sporangia suspension. Area under the curve (AUC) measurements demonstrate significantly faster sporulation of heterokaryon derivative B compared with homokaryon derivative I ($p < 0.01$, $n = 4$). Error bars depict standard error over four replicates. **b** Growth curves of six heterokaryotic isolates (black lines) versus four homokaryotic isolates (red lines) on differential lettuce lines NunDM15 (*Dm15*) and R4T57D (*Dm4*) demonstrating that viable homokaryons sporulate faster on selective hosts than heterokaryons. Measurements were taken 7, 11, 15, and 21 days post inoculation (dpi) and sporulation was measure on 20 cotyledons per observation. Source data for all panels are provided in the Source Data file

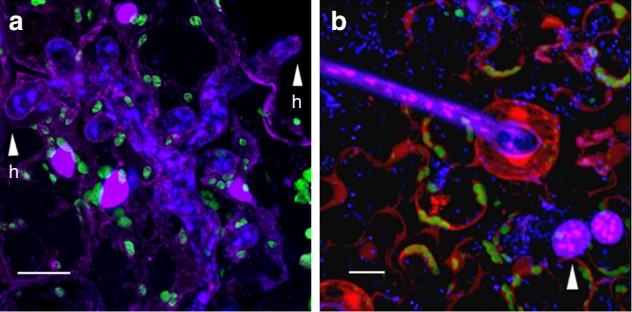

**Fig. 8** The multinucleate architecture of *B. lactucae*. Lettuce cotyledons infected with *B. lactucae* stained with 4′,6-diamidino-2-phenylindole (DAPI) to render nuclear DNA fluorescent. **a** Densely multinucleate coenocytic mycelium growing between spongy mesophyll cells of a non-transgenic lettuce cotyledon five days post infection (dpi), prior to sporulation. Two of several multinucleate haustoria that have invaginated the host plasmalemma are indicated (h). The larger plant nuclei fluoresce purple. Auto-fluorescent chloroplasts are visualized as green. **b** Infected lettuce cotyledon stably expressing DsRED stained seven dpi at the onset of sporulation. The multinucleate stem of a sporangiophore is visible exiting a stoma. Two multinucleate spores are visible on the cotyledon surface (arrowed). Small DAPI-stained bacterial cells are also visible. Scale bars in each represent 15 μm

## Discussion

Heterokaryosis in *B. lactucae* has phenotypic consequences, as well as implications for interpretation of tests for virulence phenotype. Homokaryotic derivatives G, H, and I of heterokaryotic isolate C98O622b are race Bl:5-US and derivative J has a novel virulence phenotype. The original field isolate C98O622b and heterokaryotic derivatives are race Bl:6-US, indicating that two phenotypically distinct isolates may combine to create a new phenotype when characterized on individual resistance genes, although such somatic hybrids may not be able to overcome combinations of these resistance genes in a single cultivar. Therefore, (a)virulence phenotypes on monogenic differentials are not necessarily a good predictor of virulence when heterokaryons are tested. The instability of heterokaryosis may enable a successful infection and proliferation of individual nuclear components. Furthermore, there is no a priori reason why coenocytic mycelia are limited to having only two nuclear types. Multiple rounds of somatic fusion are possible if favored by selection. Allele frequency plots are consistent with some isolates having more than two distinct nuclei (e.g., isolate C04O1017; Fig. 4c). Heterokaryotic isolates of *B. lactucae* were detected globally, over a 33-year period, from both Europe and the Western United States of America (Supplementary Table 7). Of the 31 isolates sequenced to over 50x coverage, 18 were determined to have allele frequencies deviating from that of a homokaryon. Neither the geographic location of sampling nor year of sampling correlated with heterokaryosis. Therefore, heterokaryotic isolates should be considered as exhibiting somatic hybrid vigor and selection for heterosis in *B. lactucae* as acting on populations of nuclei within a coenocytic mycelium (Fig. 8) rather than on individual isolates.

Heterokaryosis may be a common phenomenon in other oomycetes that has yet to be investigated extensively. Flow cytometry revealed heterogeneous nuclear sizes in mycelia of *P. infestans*, although stability over multiple asexual generations was not reported[13]. Somatic fusion may be a route to allopolyploidy; inter-species somatic fusion could result in transient heterokaryosis before nuclear fusion to form a somatic allopolyploid circumventing gametic incompatibility. Somatic sporangial fusions have also been reported in a basal holocarpic oomycete[43],

demonstrating the possibility of widespread heterokaryosis within the Oomycota. Heterokaryosis may be more prevalent in non-zoospore producing oomycetes. Production of zoospores with single nuclei during the asexual cycle, exhibited by most oomycetes, breaks the heterokaryotic state each asexual generation[15,16]. However, some downy mildews and *Phytophthora* spp. germinate directly from multinucleate sporangia, which potentially maintains the heterokaryotic state, as shown in our data. Increased fitness and phenotypic plasticity of heterokaryosis could be one of the selective forces favoring the loss of zoosporogenesis in multiple lineages of oomycete pathogens[17,18].

Heterokaryosis should be considered when implementing strategies for deployment of resistance genes. Cycles of somatic fusion to increase fitness and selection on populations of nuclei provide potentially great phenotypic plasticity without mutation. This could result in rapid changes in pathogen populations in response to changes in host genotypes or fungicide use. Comprehensive knowledge of the prevalence and virulence phenotypes of homokaryotic and heterokaryotic isolates, as well as the population dynamics are necessary to predict the evolutionary potential of a pathogen population.

## Methods

**Isolation culturing and DNA extraction.** *Bremia lactucae* isolate SF5 has been reported previously[14,23,24]. Additional field isolates surveyed in this study were either collected from California/Arizona between 1982 and 2015 or were supplied by Diederik Smilde (Naktuinbouw, The Netherlands; Supplementary Table 7). Sexual progeny of SF5 × C82P24 were generated by co-inoculating isolates onto cv. Cobham Green to generate oospores. Oospores were matured for several weeks in decaying plant tissue before maceration to produce a suspension. Sexual progeny were isolated by growing cv. Cobham Green in a dilute suspension of oospores. Oospores were titrated by assessing the number of infected seedlings resulting from a serial dilution of the oospore suspension. The suspension was then diluted so that on average a single seedling would be infected per culture box. Some sexual progeny sequenced have been reported previously[23,24]. Single-spore isolates were derived from cotyledons that had been sporulating asexually for 1 to 2 days typically 6 to 7 days post-infection (dpi). A single cotyledon was run over a 0.5% water agar plate until clean of spores. Single sporangia were located under a dissection microscope, pulled off the agar using pipette tips, and ejected onto fresh, 7-day-old cotyledons of cv. Green Towers that had been dampened with a drop of deionized water. Plates were incubated at 15 °C and checked for sporulation after 5 dpi. Successful single-spore infections were transferred to cv. Green Towers seedlings and maintained thereon. The virulence phenotype was determined by inoculation onto the IBEB EU-B standardized differential set (http://www.worldseed.org/wp-content/uploads/2016/05/Table-1_IBEB.pdf) and observed for sporulation at 7, 11, 15, and 21 dpi (Source Data). Fitness was determined by measuring the growth rate of *B. lactucae* of four replicates of four isolates on 20 cotyledons at 3, 5, 6, 7, and 9 days post-inoculation (dpi) on cv. Green Towers by qPCR to quantify *B. lactucae* B-tubulin (BlBtub-F: 5'-ACATACTCCGTGTGCC CTTC-3' and BlBtub-R: 5'-TCATCAGCGTTTTCAACGAG-3') relative to *L. sativa Actin* (LsActin-F: 5'-ATTACCGCTTTAGCCCCGAG-3' and LsActin-R: 5'-GCTG GAAAGTGCTGAGGGAT-3'; Source Data). The area under the curve was calculated for each replicate and significance tested using a two-tailed t-test with Holm adjustment. Additional virulence tests of heterokaryons were performed on an F[1] hybrid of NumDm15 and R4T57D, which confer resistance phenotypes Dm15 and Dm4, respectively. Microscopy was performed on ~2-week-old seedlings of lettuce cv. Green Towers, 5 dpi with *B. lactucae* isolate C16C1909 (Figs. 8a) or ~2-week-old seedlings of lettuce cv. Cobham Green homozygous for the AtUBI:: dsRED transgene, 7 dpi with *B. lactucae* isolate C98O622b (Fig. 8b). Figure 8a was captured with a Leica TCS SP8 STED 3× inverted confocal microscope using a 40× water immersion objective. Image deconvolution was performed using Huygens Professional (https://svi.nl/Huygens-Professional) and a z-projection of stacked images was made in Fiji[44]. Figure 8b was captured using a Zeiss LSM 710 laser scanning confocal microscope using a 40× water immersion objective. Z stacks were processed and combined into a single image using the ZEN Black software. Spore pellets of all isolates sequenced were obtained by washing sporangia from infected lettuce cotyledons in sterile water. Spore suspensions were concentrated by centrifugation in 15 ml tubes, resuspended, transferred to microfuge tubes, pelleted, and stored at −80 °C until DNA extraction. For DNA extraction sporangia were vortexed for two minutes in a microcentrifuge tube with approximately 200 μl of Rainex-treated beads and 0.5 ml of 2× extraction buffer (100 mM Tris-HCl pH 8.0, 1.4 M NaCl, 20 mM EDTA, 2% [wt/vol] cetyltrimethylammonium bromide, and B-mercaptoethanol at 20 μl/ml) and transferred to a fresh 2 ml tube. Material was treated with RNase (20 μl/ml) at 65 °C for 30 min. An equal volume of 1:1 phenol/chloroform was added, mixed, and centrifuged at maximum speed (5200×*g*) for

15 min. The aqueous phase was retained, mixed with 24:1 chloroform/isoamyl alcohol and again centrifuged as maximum speed for 15 min. The aqueous phase was mixed with 0.7 volumes of isopropanol and DNA precipitated at −20 °C for one hour. This was pelleted by centrifuging at maximum speed for 30 min, washed with 70% ethanol, dried, and suspended in 10 mM Tris-HCl. Quantity and quality of DNA was determined by spectrometry, as well as estimated by TAE gel electrophoresis.

**Library preparation and sequencing.** Paired-end (300 bp fragments) and mate-pair (2, 5, 7, and 9-Kb) libraries were prepared using Illumina (San Diego, CA), NEB (Ipswich, MA), and Enzymatics (Beverly, MA) reagents following the manufacturers' protocols. RNAseq libraries were constructed from cotyledons of cv. Cobham Green infected with isolate SF5 through PolyA mRNA enrichment, reverse transcription with dNTP and 2nd-strand cDNA synthesis with dUTP. cDNA was sonicated with a Covaris S220 following the manufacturer's recommendations to achieve 150-bp fragments prior to end repair and adapter ligation. Subsequent size selection and purification were performed using 1× Agencourt AMPure beads XP (Beckman Coulter, Brea, CA). The dUTP strand was then digested and libraries were amplified enriching for the first strand[45]. Synthetic long reads were generated by Moleculo (now Illumina) from barcoded libraries. Libraries were sequenced by the DNA Technologies Core at the UC Davis Genome Center (http://genomecenter.ucdavis.edu) on either a HiSeq 2500 or 4000.

The random-shear BAC library was constructed by Lucigen Corporation (Middleton, WI); this provided 10,000 BAC clones with a mean insert size of 100 kb. Sanger sequencing of BAC ends was performed by the Genome Institute at Washington University (St. Louis, MO) and generated sequences averaging 700 bp in length. A fosmid library consisting of over eight million clones with a mean insert size of 40 Kb was generated by Lucigen Corporation and end-sequenced on an Illumina MiSeq. Two SMRTbell[TM] libraries with mean insert sizes of 3 Kb and 10 Kb were constructed and sequenced by Pacific Biosciences. Hi-C libraries were produced by Dovetail Genomics. Coverage and percent of each read set mapping to the SF5 assembly are available as Supplementary Table 8.

**Flow cytometry.** Flow cytometry of select isolates was performed on sporulating cotyledons 7 dpi. For each measurement, two sporulating cotyledons were mixed with 1 cm² of young leaf tissue from *Oryza sativa* cv. Kitaake (2 C = 867 Mb), which was sufficiently different from the genome size of *B. lactucae* (2 C and 4 C) for use as the internal reference. The *O. sativa* 2 C DNA content was determined by calibrating against nuclei from flower buds of *Arabidopsis thaliana* Col-0, which has a known absolute DNA content of 2 C = 314 Mb[46]. Nuclei extraction and staining with propidium iodide was done using the CyStain PI Absolute P kit (Sysmex, Lincolnshire, IL). Flow cytometry was done on a BD FACScan (Becton Dickinson, East Rutherford, NJ). For each measurement, 10,000 nuclei were assessed, and each isolate was measured three times. Lettuce nuclei are ~3× larger than rice nuclei and did not interfere with the measurements. Data was analyzed using FlowJo (Ashland, OR). Total nuclear DNA content was averaged over all replicates. Means and standard deviations were calculated from the average nuclear content of each isolate. Haploid genome size was calculated by halving the mean across all isolates.

**De novo assembly assessment and annotation.** Multiple assembly approaches were tried using a variety of templates. Ultimately, the genome of isolate SF5 was assembled using a hybrid approach using several types of sequences (Supplementary Fig. 10). Moleculo reads were assembled using Celera[47] and further scaffolded using mate-pair, fosmid-end, BAC-end, and PacBio data, utilizing first SSPACE v3.0[48] followed by AHA[49]. A consensus assembly was obtained by removing the second haplotype using HaploMerger2[50] and further scaffolded and gap-filled[48,51]. Misjoins were iteratively detected and broken using REAPR v1.0.18 in 'aggressive mode'[52] in combination with SSPACE[48] and GapFiller[51]. Mitochondrial sequences were detected by BLASTn v2.2.28 and removed before final scaffolding and gap-filling[48,51]. Hi–C scaffolding was performed by Dovetail Genomics using their Hi-Rise pipeline to infer breaks and joins. One putative effector gene was masked by Ns in the assembly because it was determined by read coverage to be erroneously duplicated multiple times.

The quality of the assembly was assessed in multiple ways. Assembly completeness and duplication was measured by BUSCO v2, protist ensemble library db9[29], and KAT v2.4.1[53]. Nucleotide collinearity between *B. lactucae* and *Phytophthora sojae* was inferred using Promer v3.06 (-l 30) and visualized using Symap v4.2[54] set with a required minimum of 5 dots. Phylogenetic analysis was performed on nucleotide and amino acid sequences of single-copy proteins predicted by BUSCO; 18 sequences from *B. lactucae* that were also present as single copies in assemblies of all 20 *Phytophthora* spp., *Plasmopara* spp., *H. arabidopsidis*, *S. graminicola*, *Pseudoperonospora cubensis*, and *P. effusa* surveyed were aligned independently with MAFFT v7.245[55], concatenated into single sequences for each species/isolate, and phylogenetically tested with RAxML v8.0.26, run with 1000 bootstraps[56]. Alignments and trees are provided as Source Data.

A de novo transcriptome assembly was generated by mapping reads with BWA-MEM v0.7.12[57] to a combined reference of the *B. lactucae* de novo assembly and *L. sativa* assembly[58]. Reads that mapped to the *B. lactucae* assembly were assembled

with Trinity v2.2.0[59] and filtered with BLAST to ensure that the transcripts belonged to *B. lactucae*. Transcripts were translated with TransDecoder v3.0.0[60].

Primary annotation was performed using MAKER v2.31.8[61]. The RNAseq assembly was first used to predict proteins from the SF5 assembly without hidden Markov models (HMMs). These protein models were then filtered and used to produce HMMs with SNAP v2006-07-28[62], which in turn were used with MAKER for ab initio gene model predictions. This process was repeated (bootstrapped) six times. The optimal run was selected based on hit scores from BLAST[63] to the Oomycete training protein sequences, % orthology detected with the Oomycete training database as detected by OrthoFinder v2.2.1[64] and average e-value of Pfam domains[65] detected with InterProScan[66] with a value under $1e^{-5}$. Additional candidate effectors were predicted through regular expression string searches (RxLR and EER) and HMMs (WY and CRN) of all open reading frames (ORFs) over 80 residues long[17]. A previously published RxLR-EER HMM[67] was also applied, though it failed to define additional candidate effectors compared with regular expression string searches. These gene models were filtered and prepared for submission to NCBI using GAG v2.0-rc.1[68]. Genome wide transcriptional support was detected by mapping RNAseq reads back to the annotated genome with STAR v2.6.0[69] outputting the read counts per gene table. Assembly of putative effectors in the transcriptome was inferred by >=95% tBLASTn identity and $<1e^{-75}$ e-value scored between the protein and the transcript. Absence of genes encoding domains linked to zoosporogenesis and biotrophy was performed by identifying Pfam[65] domains, identified by InterProScan v5.30[66], linked to these process that were under-represented in non-flagellate/all downy mildews. Expected and observed distributions were tested with Chi-square, and p-values interpreted with Bonferroni adjustment. The analysis used the protein sequences of *Peronospora effusa*, *Pe. tabacina*, *Hyaloperonospora arabidopsidis*, *Plasmopara halstedii*, *Phytophthora infestans*, *Ph. sojae*, and *Ph. ramorum*. All proteins containing these underrepresented domains were identified and orthogroups, defined on the entire protein set by OrthoFinder v2.2.1[64], containing these proteins were visualized[17]. This dataset was also used to identify the total number of *B. lactucae* proteins with orthology to other oomycetes. Genomic coverage of gene models was calculated with BEDtools2 v2.25.0 multicov[70], multiplying the result by the read length (101 bp), and dividing by the length of the gene. Comparative annotation analysis was undertaken by downloading GFF files of annotated oomycetes from FungiDB[71] and using GAG[68] to obtain summary statistics of annotations.

**Repeat analysis.** Repeat libraries were produced independently from RepeatModeler v1.0.8 (http://www.repeatmasker.org/) and a LTRharvest v1.5.7[72]/LTRdigest v1.5.7[73]. Provisional LTRs were identified as being separated by 1–40 Kb with LTRharvest. LTRdigest was used to identify complete LTR-RTs that were then annotated by similarity to elements in TREP. Elements containing sequences annotated as genes were removed. These libraries were combined and run through RepeatMasker v4.0.6 (http://www.repeatmasker.org/). Coverage of each non-overlapping masked repeat region was calculated with BEDTools2 v2.25.0 multicov[70] using the coordinates of the repeat elements and the BAM file generated by mapping SF5 reads back to the assembly with BWA-MEM v0.7.12[57].

Divergence of LTRs for *B. lactucae* and additional oomycete assemblies was calculated by running LTRharvest[72] and LTRdigest[73] as above. Internal domains of annotated LTR-RTs were clustered with VMatch (http://www.vmatch.de/), followed by alignment of 3′ and 5′ LTRs with Clustal-O[74]. Too few internal domains were detected for *P. halstedii* to allow clustering, so it was excluded from this analysis. Divergence between aligned 3′ and 5′ LTRs was calculated with BaseML and PAML[75] and plotted using R base packages. Divergence between LTR pairs calculated for each species are provided (Source Data). LTR frequency and percentage of the assembly of each species/isolate masked was plotted with ggplot2[76] (Source Data). These predictions were not filtered for overlaps with gene annotations as multiple assemblies analyzed do not have publicly available annotations (Supplementary Table 3).

**Analyses of additional read sets.** Whole-genome sequencing data of additional oomycetes were downloaded from NCBI SRA (Supplementary Table 9) and converted to fastq files using the SRA-toolkit. Heterozygosity was calculated with GenomeScope[77]. Isolates that did not fit a diploid model were excluded from the analysis (Source Data).

Paired-end reads of all sequenced *B. lactucae* isolates were trimmed and adapter-filtered using BBMap (https://sourceforge.net/projects/bbmap/), filtered for reads of a bacterial origin by mapping to a database of all bacteria assemblies on NCBI, and mapped to the final reference assembly of SF5 using BWA-MEM v0.7.12[57]. Alternative allele frequency plots were generated by filtering BAM files for quality (>25) then generating pileups using SAMtools mpileup v0.1.18[78], followed by BCFtools v0.1.19[79] to convert to human-readable format. Bash was used to parse the files and generate the frequency of the alternative allele for every SNP that was covered by >50 reads and had an allele frequency between 0.2 to 0.8. In some instances, this frequency filter was removed to investigate the full spectrum of peaks. Bar charts were plotted with the R base package. Intersections of SNPs common to heterokaryotic and homokaryotic derivatives were obtained with BEDTools2 v2.25.0 intersect[70].

Kinship analysis was performed on progeny and derivatives sequenced to a depth greater than 10x. Reads were trimmed, filtered, and mapped as above. Multi-sample pileups were obtained with SAMtools mpileup v0.1.18[78], made human readable using BCFtools v0.1.19[79], and pairwise kinship was calculated using VCFtools v0.1.14 with the relatedness2 flag[80]. The two-column table output was transformed into a matrix using bash, and conditional formatting was used to visualize relationships. Raw matrices of these analyses are available (Source Data).

**Reporting summary.** Further information on research design is available in the Nature Research Reporting Summary linked to this article.

## Data availability

All sequence data are available at NCBI under the following BioProjects: PRJNA387017 *Bremia lactucae* asexual single spore progeny WGS. PRJNA387192 *Bremia lactucae* diversity panel WGS. PRJNA387454 *Bremia lactucae* SF5 × C82P24 progeny WGS. PRJNA387613 *Bremia lactucae* whole genome sequencing and de novo assembly. PRJNA523226 RNAseq of lettuce infected with *Bremia lactucae*. Isolates contained within each BioProject are detailed in Supplementary Table 8. Additional data used in this study are available in Supplementary Table 9. The genome assembly and annotation are available under accession GCA_004359215.1. All other relevant data is available upon request. The source data underlying Figs. 1c, 2, 3, 5, 6a, and 7 are provided as a Source Data file.

## Code availability

All software is described and cited in the article. A workflow summary of the assembly is provided (Supplementary Fig. 10a).

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

## Acknowledgements

We thank Melodie Najarro (UC Davis) for preparing *B. lactucae* cultures and Dave Tricoli for the transgenic DsRED lettuce line used for microscopy. We thank D. Smilde for supplying European (BL) isolates for sequencing. We thank Dovetail Genomics for Hi-C analysis, Dr. C.-C. Wu at Lucigen for construction of the BAC library, and the staff of the DNA Technologies Core of the UC Davis Genome Center for their sequencing efforts. This project was supported by the University of California Davis Flow Cytometry Shared Resource Laboratory with funding from the NCI P30 CA093373 (Cancer Center), and NIH NCRR C06-RR12088, S10 OD018223, S10 RR12964, and S10 RR 026825 grants and with technical assistance from Ms. Bridget McLaughlin and Mr. Jonathan Van Dyke. The work was supported by the NSF/USDA Microbial Sequencing Program award # 2009-65109-05925 and the Novozymes Inc. Endowed Chair in Genomics to R.M.

## Author contributions

K.F. performed the assembly, annotation, and inter-comparative and intra-comparative genomics, as well as drafted the manuscript. R.J.G. performed phenotyping of isolates, culturing, generation of the genetic cross, and DNA and RNA extractions. L.B. performed flow cytometry and confocal microscopy. A.K. performed culturing, DNA extractions, phenotyping of isolates, and obtained asexual single spore derivatives. K.W. prepared the qPCR investigation. L.Z., K.C., and J.W. performed culturing and DNA extractions. S.R. generated the first assembly. C.T. performed culturing, phenotyping of isolates, generation of the genetic cross, and DNA extractions. R.M. supervised and conceptualized the project and made significant contributions to all drafts. All authors contributed to the final manuscript and approved the submission.

## Additional information

**Competing interests:** The authors declare no competing interests.

