## [Peer Review File · Nature Communications]

Reviewers' comments:

Reviewer #1 (Remarks to the Author):

The paper by Fletcher and colleagues represents an important contribution to the current literature on the biology of Oomycetes, as it reveals important new aspects of their genomes, and uncover biologically relevant information regarding their genetics; particularly the evolutionary relevance of heterokaryosis.

I have a number of comments that I believe should be addressed:

Line number 91 :Results and Discussion in Genome assembly. The authors have used the PacBio sequencing as well, which are error prone. It's presently not documented in text how the sequencing errors have been resolved prior to the assembly? How were the reads polished?

Line 126: 26 to 46% should be written instead of twenty-six to 46%

Line number 114: did authors check for the telomere repeats in the high quality assembly?

Line number 768 : spell check of Phytophthora

Line number 135: what are the 18 proteins chose from BUSCO to build the phylogeny? On what basis have these been chosen? What is the level of similarity with each other at least among the mildew groups ?

Line number 144: Author mentioned that 9781 gene models are predicted. I suggest authors can try using RNAseq data as a hints or use the reference gene models to train and retrain their genes , can compare the results.

Line number 144: From the predicted gene, how many gene models are valided with with RNAseq data? Oomycete genomes typically harbor around 12,000 genes

line number 147: what is the 20.kb gene code for? Can the authors provide some information regarding its biological function?

line number 162: From the identified RXLRs from genome and transcriptome mentioned in Table 1, is it possible that authors can validate a few effectors experimentally?

Methods: (Library preparation and sequencing)

Line number 357: Coverage information and Total number of reads are lacking, since various sequencing technologies were used for this, a table representing a sequencing technology would be helpful for coverage and total number of reads.

Line number 425: Please clarify this: multiplying the result by the read length (101 bp) . Is it average read length of Genomic or RNAseq reads?

Figure 3: why is the support values between the bigger branches so low- e.g. values of 43 at the node separating Phytophthora agathidicida and Pernospora showing values of 55 and 66? Is this consistent with the present knowledge of the Oomycete phylogeny? If not, can the author indicate why this is not

the case?

- Hi-c sequencing was also used to sequence the genome, and to identify the chromatin interaction across the genome. I am curious to know how this helps in availing high quality genome assembly? Did the authors made any effort to study the chromatin interactions or how this technology helped in getting high quality genome assembly?

Reviewer #2 (Remarks to the Author):

This paper describes the sequencing of the genome of *Bremia lactucae*, a plant pathogen, and the analysis of the nuclear state of isolates. The major biological conclusion is that many isolates of *Bremia* exist in a heterozygous nuclear state, which may be related to the fact that unlike many of its relatives this species has lost the zoospore stage, which normally would serve to purify single nuclei. The results are very interesting, but there is room for improvement in several areas.

The major area for enhancement concerns the scope of analysis of the heterokaryons. This is a very interesting story, but more details are needed. We are shown many graphs of California and European isolates, for example in Supplementary Fig. 4. It would help if we knew more about the population structure of *Bremia*, and the origins of the isolates. Could the SNP data (or a subset of the data) be used to generate phylogenetic trees to indicate if the heterokaryons have diverse origins? Was the sampling of isolates representative of the species, or did the mode of sampling lead to a bias (were the isolates selected because they broke resistance, for example?). Knowing the dates of isolation could also be useful. Are older isolates more likely to be homokaryotic? Did the deployment of resistance genes drive heterokaryon formation?

I suggest that the authors include more of the allele ratio graphs (such as those in Fig. S4) in the main text; this would add to the story. Maybe this could be tied into the phylogram described above. Using space for that purpose would seem more important than including the large heterozygosity graph shown in Fig. 1. On the latter topic, while the authors state that *Bremia* has high heterozygosity (Fig. 1c), the level of heterozygosity is actually quite low compared to most species, and I don't think that heterozygosity was so high that it would hinder assembly.

Another area for improvement concerns the studies of fitness. The authors claim that heterokaryons are more fit. The number of strains analysed (four from the same parents) seems too few to draw this conclusion.

The results section describes that a range of technologies that were used to generate the assembly (PacBio, Illumina, BAC-end sequencing, Hi-C etc.). Compared to most genome projects, the efforts were quite heroic and this could be stressed more in the beginning of Results (li 91). It would be interesting in the authors could speculate which technologies were most useful. Also, this might be more obvious if the authors indicated which steps of the assembly process were represented by the graphs in Supplemental Fig. 10b. Also, the authors should check for consistency between Supplemental Fig 10a and Results, as there are some differences (such as the order of AHA and SSPACE).

It is important to ensure that the data in the paper are made available. Several NCBI Bioproject numbers are indicated in the manuscript, but it is unclear what these refer to. Do these represent the genome assembly, RNA seq reads, or the different types of DNA reads used for assembly? Will gene annotations be deposited?

Many figure legends require more details (or more careful writing) to help the reader. For example, abbreviations are often not described. I can guess what some are, but others are not as clear. A few examples: Fig. 7 GT, AUC, and ASSP in Fig. 7, the meaning of OS, and P24-1 vs. P24-2 in Fig. 1. Also, a little explanation of the colors in the Kinship analysis would also be helpful. Also, do the different spots in Fig. 1c mean different isolates? Also, Fig. 2a is described as showing "Ages" but could this be described better? Please look at each figure legend and add details as appropriate.

The description of genome content is sometimes confusing (line 109 and elsewhere). At some points genome content is described based on the assembly, but the authors indicate that about 40% of the genome is missing from the assembly. To make things more clear, perhaps some tables could include separate data columns for genome and assembly?

It is also deceiving for the authors to state (Abstract line 36) that they have generated a "high-quality, chromosome-scale, consensus assembly." It could only be considered as such if the missing 40% (mostly overcollapsed repeated DNA) is ignored. Also, where are the data to support the statement that the assembly is "chromosome-scale?" The assembly contains 22 scaffolds, but the existence of about half that number of chromosomes were suggested by prior pulsed-field gel analysis and cytological analysis. Might the scaffolds not extend through the centromeres? Along the same lines, it would be interesting if the authors could indicate the number and location of telomeres in the assembly.

The authors describe the presence of accessory chromosomes in Introduction, but this is not discussed later in the paper. Can the authors draw any conclusions about their stability and inheritance from their sequence analysis of progenies?

Li 177: The missing genes in downy mildews have already been described by the authors in their BMC Genomics paper (which focused on other species); this paper should be cited here.

Line 56: What is the evidence that these organisms are "highly variable" compared to the average species? All because traits change doesn't make them highly variable. This is a minor point, but the authors seem to be implying that the genomes foster rapid change while the polycyclic nature of the disease (many crops of spores per season) could be the explanation.

More minor issues:

The authors refer to *Bremia* spores as sporangia, but they are conidia.

I think that a careful reading by the authors is needed to catch many small errors. A few examples: line 118, which supplemental file do they refer to? li 190, zoosporogenesis is spelled incorrectly. li 96, col(l)inear. Fig S10b legend, significant(ly). Li 359, Enzymatic(S). *S. parasitica* is misspelled in Fig. 1. Also consider using Al for *Albugo* and Ap for *Aphanomyces* in this figure.

Reviewer #1 (Remarks to the Author):

The paper by Fletcher and colleagues represents an important contribution to the current literature on the biology of Oomycetes, as it reveals important new aspects of their genomes, and uncover biologically relevant information regarding their genetics; particularly the evolutionary relevance of heterokaryosis.

I have a number of comments that I believe should be addressed:

Line number 91 :Results and Discussion in Genome assembly. The authors have used the PacBio sequencing as well, which are error prone. It's presently not documented in text how the sequencing errors have been resolved prior to the assembly? How were the reads polished?

No error correction of PacBio sequences was required because they were only used in AHA to scaffold the Moleculo assembly. AHA does not add sequence to the assembly and therefore all bases in the *B. lactucae* assembly were derived from high confidence Illumina reads. The total sequence of the pre and post AHA assemblies were identical with no additional sequence that required polishing.

Line 126: 26 to 46% should be written instead of twenty-six to 46%

Change accepted.

Line number 114: did authors check for the telomere repeats in the high quality assembly?

This is an interesting point. We checked the assembly for the telomeric repeat TTTAGGG and its reverse complement. We found two >1 Mb scaffolds that had terminal telomeric sequences. Other scaffolds had telomeric-like repeats but these were not at the ends of scaffolds, which may be the biological reality. While knowledge of telomeres would be nice and warrant further analysis, it is tangential to the biological conclusions of the paper and therefore were not considered in this paper.

Line number 768 : spell check of *Phytophthora*

Corrected.

Line number 135: what are the 18 proteins chose from BUSCO to build the phylogeny? On what basis have these been chosen? What is the level of similarity with each other at least among the mildew groups ?

Added criteria for use of BUSCO protein; “ubiquitously single-copy in the species surveyed”. In addition, we have provided the alignments as Supplementary Data.

Line number 144: Author mentioned that 9781 gene models are predicted. I suggest authors can try using RNAseq data as a hints or use the reference gene models to train and retrain their genes, can compare the results.

We agree with the value of this approach and this is the strategy that we used for annotation. This has been clarified in the methods:

“This process was repeated (bootstrapped) six times⁸⁸. The optimal run was selected based on hit scores from BLAST⁸⁹ to the Oomycete training protein sequences, % orthology detected with the Oomycete training database as detected by OrthoFinder⁹⁰ and average e-value of Pfam domains⁹¹ detected with InterProScan⁹² with a value under 1e-5”

Line number 144: From the predicted gene, how many gene models are validated with RNAseq data? Oomycete genomes typically harbor around 12,000 genes

We have mapped the reads of the RNAseq data that was used to annotate the genome (infected lettuce cotyledons) back to the *B. lactucae* assembly. We found strand specific coverage for 7253/9769 (74 %) of the genes. We have now included this result in the manuscript:

“Strand specific transcriptional support was detected for 74 % of the genes.”

line number 147: what is the 20.kb gene code for? Can the authors provide some information regarding its biological function?

We added that it is conserved in several oomycetes:

“The larger genes were conserved with other oomycete gene models”.

This includes *Phytophthora cactorum*, *P. parasitica*, *Saprolegnia diclina*, *Albugo laibachii* and *A. candida*. However, no clear biological functional inferences can be made from its sequence.

line number 162: From the identified RXLRs from genome and transcriptome mentioned in Table 1, is it possible that authors can validate a few effectors experimentally?

We agree that this is biologically relevant information and that it should be made publicly available. Publication of this assembly enables comparative work against other oomycetes and allow other labs to mine this resource for effectors/proteins. Experimental validation of effectors is outside of the scope of this paper; functional analysis of effectors is underway and will be the subject of another major study.

Methods: (Library preparation and sequencing)

Line number 357: Coverage information and Total number of reads are lacking, since various sequencing technologies were used for this, a table representing a sequencing technology would be helpful for coverage and total number of reads.

This is a good suggestion. The requested data is now provided in Supplemental Table 7.

Line number 425: Please clarify this: multiplying the result by the read length (101 bp). Is it average read length of Genomic or RNAseq reads?

Genomic. This has been clarified:

“Genomic coverage of gene models was calculated with BEDtools2 v2.25.0 multicov⁹⁵, multiplying the result by the read length (101 bp), and dividing by the length of the gene.”

Figure 3: why is the support values between the bigger branches so low- e.g. values of 43 at the node separating *Phytophthora agathidicida* and *Pernospora* showing values of 55 and 66? Is this consistent with the present knowledge of the Oomycete phylogeny? If not, can the author indicate why this is not the case?

We agree that this is not as strongly supported as we would like and it warranted further investigation; we now present the tree with the bootstrap values for both amino acid and nucleotide based trees. The latter analysis provides much stronger support for the same tree topology and therefore a polyphyletic origin for the downy mildews. This tree is consistent with other studies that used different datasets (Fletcher *et al.* 2018, Bourett *et al.* 2018, and McCarthy and Fitzpatrick 2017). We have revised the text to read:

“This is consistent with the biotrophic downy mildews having evolved at least twice from hemi-biotrophic *Phytophthora*-like ancestors. These results are similar to previous, less extensive studies⁴⁷⁻⁵⁰. Additional genome sequencing of both biotrophic and hemi-biotrophic Peronosporales species will enable further clarification with regards to the origin(s) of the downy mildews.”

- Hi-c sequencing was also used to sequence the genome, and to identify the chromatin interaction across the genome. I am curious to know how this helps in availing high quality genome assembly? Did the authors made any effort to study the chromatin interactions or how this technology helped in getting high quality genome assembly?

We used Hi-C data to greatly increase contiguity of the assembly as stated. We did not use it to study chromatin interactions (such as inter-chromosomal interactions or TAD analysis). Such studies, while interesting, require considerable replication to be informative and are beyond the scope of this paper.

“Subsequently, 87.9 Mb (96.5%) of the assembled sequence was placed into 22 scaffolds over 1 Mb using Hi-C; these totaled 112 Mb including gaps. The resultant assembly was highly collinear and comparable to the highly contiguous v3.0 assembly of *Phytophthora sojae* (45), which cross-validates the high quality of both assemblies (Fig. 1b).”

Reviewer #2 (Remarks to the Author):

This paper describes the sequencing of the genome of *Bremia lactucae*, a plant pathogen, and the analysis of the nuclear state of isolates. The major biological conclusion is that many isolates of *Bremia* exist in a heterozygous nuclear state, which may be related to the fact that unlike many of its relatives this species has lost the zoospore stage, which normally would serve to purify single nuclei. The results are very interesting, but there is room for improvement in several areas.

The major area for enhancement concerns the scope of analysis of the heterokaryons. This is a very interesting story, but more details are needed. We are shown many graphs of California and European isolates, for example in Supplementary Fig. 4. It would help if we knew more about the population structure of *Bremia*, and the origins of the isolates. Could the SNP data (or a subset of the data) be used to generate phylogenetic trees to indicate if the heterokaryons have diverse origins?

The population structure of *Bremia* is an interesting question. However, it is complex and we are still unravelling it using more isolates that have been sequenced at low coverage (with insufficient depth for determining heterokaryosis and so cannot be included in this study). It would be premature to present a phylogenetic analysis. The 31 isolates represent diverse temporal and geographic origins. This information was available in supplementary figure 4 but now has been added as supplementary table 6. We have emphasized in the text that heterokaryons were detected in both young and old European and California populations.

“Heterokaryotic isolates of *B. lactucae* were detected globally, over a 33-year period, from both Europe and the Western United States of America (Supplementary Table 6). Of the 31 isolates sequenced to over 50x coverage, 18 were determined to have allele frequencies deviating from that of a homokaryon. Neither the geographic location of sampling nor year of sampling correlated with heterokaryosis.”

Was the sampling of isolates representative of the species, or did the mode of sampling lead to a bias (were the isolates selected because they broke resistance, for example?).

Isolates were part of our global collection and have been accumulated over many years. There was no bias as to these isolates breaking a resistance gene; they were not collected from the same geographic location or in a specific year.

Knowing the dates of isolation could also be useful. Are older isolates more likely to be homokaryotic? Did the deployment of resistance genes drive heterokaryon formation?

This is now provided in supplementary table 6. There was no correlation with age of isolate and heterokaryosis.

I suggest that the authors include more of the allele ratio graphs (such as those in Fig. S4) in the main text; this would add to the story. Maybe this could be tied into the phylogram described above. Using space for that purpose would seem more important than including the large heterozygosity graph shown in Fig. 1.

We disagree. Fig. 4 is sufficiently illustrative of the diversity of allele ratio graphs and we state in the text how many isolates have each type. Fig. S4 appropriately provides the whole dataset.

On the latter topic, while the authors state that *Bremia* has high heterozygosity (Fig. 1c), the level of heterozygosity is actually quite low compared to most species, and I don't think that heterozygosity was so high that it would hinder assembly.

The assembled isolate SF5 of *B. lactucae* has 1.17% heterozygosity; this is much higher than most animal, plant, and oomycete species and which equals an average of a SNP every 85 bp through the genome, i.e. a polymorphism in the majority of reads. Such a level of heterozygosity confounds graph-based assembly methods resulting in a large amount of redundant assembly of haplotypes, which had to be removed post assembly. Organisms with lower heterozygosity are easier to assemble as we recently reported for *Peronospora effusa* (Fletcher *et al.* 2018 BMC Genomics).

Another area for improvement concerns the studies of fitness. The authors claim that heterokaryons are more fit. The number of strains analysed (four from the same parents) seems too few to draw this conclusion.

We agree that additional data would be desirable. These are difficult experiments to do. We cannot currently combine homokaryotic isolates and create a heterokaryon and then determine their fitness. We report the only data that we have been able to generate based on comparing homokaryotic single spore derivatives to their heterokaryotic sibs. Furthermore, we added that randomly selected isolates from the field were more commonly heterokaryotic than not. We have revised the text to acknowledge the limitations of the current data. The current data are consistent with *B. lactucae* gaining an advantage by being heterokaryotic. This is an important point and has many implications to candidate effector finding, haplotype resolution and field screening that warrants further investigation in this and other species.

The results section describes that a range of technologies that were used to generate the assembly (PacBio, Illumina, BAC-end sequencing, Hi-C etc.). Compared to most genome projects, the efforts were quite heroic and this could be stressed more in the beginning of Results (li 91).

We thank the reviewer for this comment – it was a major effort; however, we are satisfied with how the assembly is currently reported.

It would be interesting if the authors could speculate which technologies were most useful. Also, this might be more obvious if the authors indicated which steps of the assembly process were represented by the graphs in Supplemental Fig. 10b. Also, the authors should check for consistency between Supplemental Fig 10a and Results, as there are some differences (such as the order of AHA and SSPACE). We clarified the methods section (which presumably the reviewer was referring to rather than the results section) and provided further details on the order that software was applied.

“Moleculo reads were assembled using Celera⁷¹ and further scaffolded using mate-pair, fosmid-end, BAC-end, and PacBio data utilizing first SSPACE v3.0⁷² followed by AHA⁷³. Gaps filled with PacBio reads were not polished to ensure that repeat sequences were not erroneously polished to higher identity. A consensus assembly was obtained by removing the second haplotype using Haplomerger2⁷⁴ and further scaffolded and gap-filled^{72,75}. Misjoins were iteratively detected and broken using REAPR v1.0.18 in ‘aggressive mode’⁷⁶ in combination with SSPACE⁷² and GapFiller⁷⁵.”

It is important to ensure that the data in the paper are made available. Several NCBI Bioproject numbers are indicated in the manuscript, but it is unclear what these refer to. Do these represent the genome assembly, RNA seq reads, or the different types of DNA reads used for assembly? Will gene annotations be deposited?

We agree. All the data is being made available.

Our Bioprojects include as detailed in the text:

PRJNA387017 *Bremia lactucae* asexual single spore progeny WGS.

PRJNA387192 *Bremia lactucae* diversity panel WGS.

PRJNA387454 *Bremia lactucae* SF5 x C82P24 progeny WGS.

PRJNA387613 *Bremia lactucae* whole genome sequencing and de novo assembly.

PRJNA523226 RNAseq of lettuce infected with *Bremia lactucae*.

Isolates contained within each BioProject are detailed in Supplementary Table 9. Additional data used in this study are available in Supplementary Table 8.

Many figure legends require more details (or more careful writing) to help the reader. For example, abbreviations are often not described. I can guess what some are, but others are not as clear. A few examples: Fig. 7 GT, AUC, and ASSP in Fig. 7, the meaning of OS, and P24-1 vs. P24-2 in Fig. 1.

Legend was expanded and corrected.

Also, a little explanation of the colors in the Kinship analysis would also be helpful.

Added color scales to figures.

Also, do the different spots in Fig. 1c mean different isolates?

Yes; clarified in legend.

Also, Fig. 2a is described as showing "Ages" but could this be described better?

Revised "ages" to "percent divergence" in legend.

Please look at each figure legend and add details as appropriate.

Done.

The description of genome content is sometimes confusing (line 109 and elsewhere).

In response, we appropriately revised the manuscript in several places. We now consistently refer to the assembly as the sequence we have generated.

At some points genome content is described based on the assembly, but the authors indicate that about 40% of the genome is missing from the assembly. To make things more clear, perhaps some tables could include separate data columns for genome and assembly?

The data that we have is for the assembly unless otherwise stated. Every table header indicates that the data is being surveyed from the assembly. Adding a genome column would contain data inferred from the assembly and therefore would be redundant.

It is also deceiving for the authors to state (Abstract line 36) that they have generated a "high-quality, chromosome-scale, consensus assembly." It could only be considered as such if the missing 40% (mostly overcollapsed repeated DNA) is ignored.

We stand by this statement and do not ignore the collapsed portion. We believe this is a high-quality assembly because it contains nearly all of the gene space in megabase scaffolds that are highly colinear to another well assembled related species. Although the assembly has many collapsed repeats, this has not disrupted the contiguity or gene content. That we were able to resolve recently diverged repeats within the assembly supports how good the content of the assembly is. K-mer inclusion and BUSCO benchmarking clearly supports that this is a consensus assembly.

Also, where are the data to support the statement that the assembly is "chromosome-scale?" The assembly contains 22 scaffolds, but the existence of about half that number of chromosomes were suggested by prior pulsed-field gel analysis and cytological analysis. Might the scaffolds not extend through the centromeres?

We did not mean to imply that it was an end-to-end chromosomal assembly. It may well be a mixture of chromosome arms and near-whole chromosomes. We have therefore changed the wording to near-chromosome scale.

Along the same lines, it would be interesting if the authors could indicate the number and location of telomeres in the assembly.

Please see the comment above in response to reviewer 1.

The authors describe the presence of accessory chromosomes in Introduction, but this is not discussed later in the paper. Can the authors draw any conclusions about their stability and inheritance from their sequence analysis of progenies?

So far, we have not been able to assign scaffolds to the previously reported accessory chromosomes. This is currently under investigation.

Li 177: The missing genes in downy mildews have already been described by the authors in their BMC Genomics paper (which focused on other species); this paper should be cited here.

Citation added.

Line 56: What is the evidence that these organisms are "highly variable" compared to the average species? All because traits change doesn't make them highly variable. This is a minor point, but the authors seem to be implying that the genomes foster rapid change while the polycyclic nature of the disease (many crops of spores per season) could be the explanation.

High variability within the downy mildews refers to their ability to rapidly overcome resistance and fungicide treatments. We do not claim rates of variation relative to other species.

More minor issues:

The authors refer to *Bremia* spores as sporangia, but they are conidia.

Both conidia and sporangia have been used for *Bremia*. *Phytophthora* spp. and multiple downy mildews produce sporangia and release zoospores. *Bremia* and some other downy mildews have lost the ability to produce zoospores and infect after direct germination of sporangia/conidia. In addition, some *Phytophthora* spp. can germinate both directly and via zoospores. Some consider the term conidia should be restricted to the true fungi, while others consider it to be a more taxonomically generic term. Given that we are considering *Bremia* in a taxonomic context, we have retained the use of sporangia rather than conidia.

I think that a careful reading by the authors is needed to catch many small errors. A few examples: line 118, which supplemental file do they refer to? li 190, zoosporogenesis is spelled incorrectly. li 96, col(l)inear. Fig S10b legend, significant(ly). Li 359, Enzymatic(S). *S. parasitica* is misspelled in Fig. 1. Also consider using Al for *Albugo* and Ap for *Aphanomyces* in this figure.

Corrected all of the above. Thanks for catching these errors.

REVIEWERS' COMMENTS:

Reviewer #1 (Remarks to the Author):

The authors have properly addressed all my comments.

Reviewer #2 (Remarks to the Author):

This revised version of the manuscript has been improved slightly compared to the original version. It remains an impressive story of an assembly of a complex genome, although this is becoming more commonplace with the technologies that have emerged in the past few years. The description of heterokaryosis within the species is also interesting. I am also pleased to see that the authors have now deposited their data (except apparently for gene annotations) in GenBank.

I still have a major concern, however. What makes the story the most interesting is the conclusion that heterokaryosis provides "somatic hybrid vigor" (as shown in the title) and that heterokaryons have "increased fitness compared to homokaryotic derivatives" as stated in the abstract. But I do not believe that the data support this. Data used to support this claim are in Fig. 7. The authors state that "the heterokaryotic derivatives grew faster than either homokaryotic derivative." (li 278). This conclusion was drawn from only four strains, derived from but a single parent. Moreover, as shown in the top graph in Fig. 7 the rate of growth of heterokaryon A is not significantly different from homokaryon J. Admittedly heterokaryon B grew faster than homokaryon I. Nevertheless by conventional criteria the difference between the two heterokaryons and two homokaryons is not significantly different ($P=0.25$). Certainly the impact of heterokaryosis may be significant, but the data do not support higher fitness. This is more of a conclusion based on the false logic that if heterokaryons exist, they must be more fit, as well as the theory that if heterokaryons break down in the field then a homokaryon may be able to escape recognition by an R gene. Population genetics data might support whether the latter has actually occurred. I agree with the authors that adding population genetics data would be a lot of work, but it might add weight to their conclusion about fitness for which evidence appears currently to be lacking.

In their Response to Reviewers document, the authors agree that additional data would be desirable, and state that the experiment is a difficult one to do. But if the data do not support the claim of increased fitness, why is the claim still made in the title and abstract? It would be more appropriate to limit the somatic hybrid vigor theory in Discussion. The authors also state in Response to Reviewers that " We have revised the text to acknowledge the limitations of the current data" but on the assumption that the yellow highlighted text represent the changes, I see no evidence of this in the revision.

Less important issues:

I had suggested that in some of the tables, that "assembly" values be replaced by "genome" values since much of the assembly was represented by overcollapsed repeats. The authors did not want to do this, but this remains an issue in Table 2, which lists the repeat content of the assembly. Presumably readers would be more interested in the content of the *Bremia* genome than of the assembly.

on line 149 the authors state that genes "ranged from 180 bp to 20.7 Kb. The larger genes were conserved with other oomycete gene models." But this could be stated more clearly: are they just talking about the one very large gene or all "large" genes?

Reviewer #1 (Remarks to the Author):

The authors have properly addressed all my comments.

We thank the reviewer for his time and critique of our manuscript.

Reviewer #2 (Remarks to the Author):

This revised version of the manuscript has been improved slightly compared to the original version. It remains an impressive story of an assembly of a complex genome, although this is becoming more commonplace with the technologies that have emerged in the past few years. The description of heterokaryosis within the species is also interesting. I am also pleased to see that the authors have now deposited their data (except apparently for gene annotations) in GenBank.

We thank the reviewer for this comment. The annotations were submitted to Genbank many months ago and released after processing during the latest round of review, so there was a delay between the release of the assembly and the annotation. They are currently available at: ftp.ncbi.nlm.nih.gov/genomes/all/GCA/004/359/215/GCA_004359215.1_BlacSF5

I still have a major concern, however. What makes the story the most interesting is the conclusion that heterokaryosis provides "somatic hybrid vigor" (as shown in the title) and that heterokaryons have "increased fitness compared to homokaryotic derivatives" as stated in the abstract. But I do not believe that the data support this. Data used to support this claim are in Fig. 7. The authors state that "the heterokaryotic derivatives grew faster than either homokaryotic derivative." (li 278). This conclusion was drawn from only four strains, derived from but a single parent. Moreover, as shown in the top graph in Fig. 7 the rate of growth of heterokaryon A is not significantly different from homokaryon J. Admittedly heterokaryon B grew faster than homokaryon I. Nevertheless by conventional criteria the difference between the two heterokaryons and two homokaryons is not significantly different ($P=0.25$). Certainly the impact of heterokaryosis may be significant, but the data do not support higher fitness. This is more of a conclusion based on the false logic that if heterokaryons exist, they must be more fit, as well as the theory that if heterokaryons break down in the field then a homokaryon may be able to escape recognition by an R gene.

Based on this concern (and below) we have scaled back our references to hybrid vigor in the main text of this manuscript and removed it from the title. We still believe that our data provides solid evidence for somatic hybrid vigor. The heterokaryon sporulated significantly better than one homokaryotic derivative and the heterokaryon was able to infect a greater range of resistant cultivars than either homokaryon. We have therefore retained this point in the discussion.

Population genetics data might support whether the latter has actually occurred. I agree with the authors that adding population genetics data would be a lot of work, but it might add weight to their conclusion about fitness for which evidence appears currently to be lacking.

We agree with the reviewer that population genetics data would strengthen the evidence but it is beyond the scope of the current paper. We show that more than half of the 31 field isolates

analyzed were heterokaryotic. We do not claim that the prevalence of heterokaryons is evidence for their increased fitness, only that heterokaryosis is common; however, given that sexual progeny are homokaryotic, there must be selection for their generation and persistence.

In their Response to Reviewers document, the authors agree that additional data would be desirable, and state that the experiment is a difficult one to do. But if the data do not support the claim of increased fitness, why is the claim still made in the title and abstract? It would be more appropriate to limit the somatic hybrid vigor theory in Discussion. The authors also state in Response to Reviewers that " We have revised the text to acknowledge the limitations of the current data" but on the assumption that the yellow highlighted text represent the changes, I see no evidence of this in the revision.

We have revised the text as described above.

Less important issues:

I had suggested that in some of the tables, that "assembly" values be replaced by "genome" values since much of the assembly was represented by overcollapsed repeats. The authors did not want to do this, but this remains an issue in Table 2, which lists the repeat content of the assembly. Presumably readers would be more interested in the content of the *Bremia* genome than of the assembly.

We understand what the reviewer is requesting; however, inferring the number of bases present as repeats in the genome from reads mapped to an assembly would be noisy and assume that the mapping and indel calling is robust. We therefore believe the most accurate strategy is to identify the repetitive regions of the assembly, which is done in what is now Table 1 (was Table 2 in the last review) and highlight that these repeats are likely collapsed as stated in the Genome Assembly section of the results.

on line 149 the authors state that genes "ranged from 180 bp to 20.7 Kb. The larger genes were conserved with other oomycete gene models." But this could be stated more clearly: are they just talking about the one very large gene or all "large" genes?

In response to this we have searched the orthology data for *Bremia lactucae*. The largest 142 proteins have orthology with other oomycetes. We now state this in the text.